

# Volatility of aerosol particles from NO₃ oxidation of various biogenic organic precursors

Emelie L. Graham[1] #, Cheng Wu[1,a] #, David M. Bell[2], Amelie Bertrand[2], Sophie L. Haslett[1], Urs Baltensperger[2], Imad El Haddad[2], Radovan Krejci[1], Ilona Riipinen[1] and Claudia Mohr[1]

[1]Department of Environmental Science (ACES) and Bolin Centre for Climate Research, Stockholm University, Stockholm, Sweden
[2]Paul Scherrer Institute, Laboratory of Atmospheric Chemistry, 5232 Villigen, Switzerland
[a]now at: Department of Chemistry and Molecular Biology, University of Gothenburg, Gothenburg, Sweden
# These authors contributed equally to this work.

*Correspondence to*: Ilona.Riipinen@aces.su.se and Claudia.Mohr@aces.su.se

## Abstract

Secondary organic aerosol (SOA) is formed through the oxidation of volatile organic compounds (VOC), which can be of both natural and anthropogenic origin. While the hydroxyl radical (OH) and ozone (O₃) are the main atmospheric oxidants during the day, the nitrate radical (NO₃) becomes more important during the night time. Yet, atmospheric nitrate chemistry has received less attention compared to OH and O₃.

The Nitrate Aerosol and Volatility Experiment (NArVE) aimed to study the NO₃-induced SOA formation and
evolution from three biogenic VOCs (BVOC), namely isoprene, α-pinene and β-caryophyllene. The volatility of aerosol particles was studied using isothermal evaporation chambers, temperature-dependent evaporation in a volatility tandem differential mobility analyzer (VTDMA), and thermal desorption in a filter inlet for gases and aerosols coupled to a chemical ionization mass spectrometer (FIGAERO-CIMS). Data from these three setups present a cohesive picture of the volatility of the SOA formed in the dark from the three biogenic precursors. Under our experimental conditions, the SOA formed from
NO₃ + α-pinene was generally more volatile than SOA from α-pinene ozonolysis, while the NO₃ oxidation of isoprene produced similar, although slightly less volatile SOA than α-pinene under our experimental conditions. β-caryophyllene reactions with NO₃ resulted in the least volatile species.

Three different parameterizations for estimating the saturation vapor pressure of the oxidation products were tested for reproducing the observed evaporation in a kinetic modelling framework. Our results show that the SOA from nitrate
oxidation of α-pinene or isoprene is dominated by low volatility organic compounds (LVOC) and semivolatile organic compounds (SVOC), while the corresponding SOA from β-caryophyllene consists primarily of extremely low volatility




organic compounds (ELVOC) and LVOC. The parameterizations yielded variable results in terms of reproducing the observed evaporation, and generally the comparisons pointed to a need for re-evaluating the treatment of the nitrate group in such parameterizations. Strategies for improving the predictive power of the volatility parameterizations, particularly in relation to

the contribution from the nitrate group, are discussed.

## 1 Introduction

Atmospheric aerosols, i.e. liquid or solid particles suspended in air, are important for the climate and human health. A large fraction of atmospheric particulate matter (PM) is secondary, formed from the condensation of atmospheric vapors after chemical reactions. Oxidation typically lowers the volatility (saturation vapor pressure) of the gas-phase precursors, resulting

in increased atmospheric particle mass through condensation, and possibly new particle formation (NPF). Secondary organic aerosol (SOA) is formed from the oxidation of volatile organic compounds (VOCs), which can be of both anthropogenic and natural origin. The atmosphere contains a plethora of VOCs making it a complex system to understand and describe.

A way to describe the volatility of complex SOA is to use the volatility basis set (VBS), where the saturation vapor pressures of individual compounds are approximated as effective saturation vapor concentrations ($C^*$, Donahue et al., 2006).

In most VBS approaches, the $C^*$ values of individual molecular compounds are grouped into discrete bins. In this way, a complex mass spectrum containing thousands of compounds can be reduced to considerably fewer data points of volatilities, simplifying the description of e.g. atmospheric aging processes in model frameworks. Volatility distributions of various SOA mixtures can be estimated experimentally based on the formation and growth or evaporation of the SOA (Berkemeier et al., 2020; Boyd et al., 2015; Karnezi et al., 2014; Lee et al., 2011; Tröstl et al., 2016), or through converting the molecular

composition of SOA observed with mass spectrometric techniques to estimates of the volatilities present in the mixture (Donahue et al., 2011; Li et al., 2016; Pankow and Asher, 2008). Direct measurements of the volatility distributions are challenging, and typically a combination of techniques is needed to probe the full range of volatilities present in atmospheric mixtures and avoid misinterpretations of data.

While the hydroxyl radical (OH) and ozone are the two dominant oxidants in the atmosphere during daytime, the

nitrate radical ($NO_3$) is the major nocturnal oxidant. It is generated in the dark by the reaction of nitrogen dioxide ($NO_2$) and $O_3$, and its reaction rates with monoterpenes are similar to OH (Ng et al., 2017). The number of studies on $NO_3$ oxidation of biogenic VOCs (BVOCs) are much fewer than those for ozonolysis and oxidation by OH, but generally show more variable SOA yields (Ng et al., 2017). Specifically, reported SOA yields for nitrate oxidation are 0–82 % for α-pinene (Bates et al., 2021; Berkemeier et al., 2020; Fry et al., 2014; Hallquist et al., 2009; Nah et al., 2016; C. Wu et al., 2021), 2–24 % for isoprene

(Ng et al., 2008; Rollins et al., 2009; C. Wu et al., 2021) , and 86 – 146 % for β-caryophyllene (Fry et al., 2014; Jaoui et al., 2013; C. Wu et al., 2021). Nitrate oxidation of BVOCs is also a major pathway for forming atmospheric organic nitrates (ONs), which have been shown to be an important component of ambient SOA (Häkkinen et al., 2012; Holzinger et al., 2010; Kiendler-Scharr et al., 2016; Lee et al., 2016; O'Brian et al., 2014). Early modelling studies of the atmospheric fate of ONs assumed





relatively high volatility, classifying all SOA (including ON) as semivolatile organic compounds (SVOC) with $C* > 10^{-0.5}$ µg
m$^{-3}$ (Chung and Seinfeld, 2002). ON volatility in the semivolatility range was also established experimentally, based on
equilibrium partitioning fits (Rollins et al., 2009). However, more recent studies, from both field and laboratory, suggest much
lower volatilities to be associated with ON (e.g. Berkemeier et al., 2020; Häkkinen et al., 2012), with some ON even being
characterized as highly oxygenated organic molecules (HOM) and falling within the category of extremely low volatility
organic compounds (ELVOC $< 10^{-4.5}$ µg m$^{-3}$) (Pullinen et al., 2020; Zhao et al., 2021). These later results are in line with a
rather large decrease in volatility associated with NO$_3$ addition of about 2.5 orders of magnitude based on the group
contribution method (see e.g. Pankow and Asher, 2008; Donahue et al., 2011). Apart from a handful of pioneering studies (e.g.
Berkemeier et al., 2020; C. Wu et al., 2021; R. Wu et al., 2021), research focusing specifically on the volatility of the nitrate-
induced biogenic SOA is scarce. In particular, work combining multiple different techniques for studying the volatility of
different chemical systems in a consistent manner is largely lacking.

This study is part of the Nitrate Aerosol Volatility Experiment (NArVE, see also Bell et al. (2021) and C. Wu et al. (2021)),
aimed to improve our understanding of nitrate-induced biogenic SOA formation. Bell et al. (2021) showed that the dark aging
of α-pinene + NO$_3$ was driven primarily by further oxidation reactions, in addition to fragmentation and dimer degradation
reactions. C. Wu et al. (2021), on the other hand, studied the transition from dark to light conditions, revealing significant
changes in the chemical composition but relatively low evaporation. In this paper, the focus is on the volatility of biogenic
SOA formed from the nitrate oxidation of α-pinene, isoprene and β-caryophyllene. This study combines three experimental
methods, namely isothermal evaporation chambers, temperature-dependent evaporation in the volatility tandem differential
mobility analyzer (VTDMA), and a chemical ionization mass spectrometer with a filter inlet for gases and aerosols
(FIGAERO-CIMS). The chemical information obtained from the FIGAERO-CIMS is used to estimate the volatility
distributions for the SOA systems, which is then used as input in a kinetic model simulating the evaporation of aerosol particles
in the isothermal evaporation chamber and the thermodenuder (TD) of the VTDMA. Additionally, experiments of α-pinene
ozonolysis were performed for comparison and to benchmark our methods against the literature on the volatility of this well-
studied system. The performance of the composition-based volatility predictions is evaluated, and the observations from the
three systems are compared to available literature data. Finally, implications for the volatility distributions of the studied
systems are discussed, together with strategies for improving the predictive power of the volatility parameterizations especially
for ONs.

## 2 Methods

### 2.1 Chamber conditions and experimental overview of the NArVE campaign

Atmospheric simulation chamber experiments of the oxidation of α-pinene, isoprene and β-caryophyllene were
performed at the Paul Scherrer Institute in an 8 m$^3$ Teflon chamber (see Platt et al., 2013), operating at 18-23 °C and 58-65%
relative humidity (RH) (see Table 1 for an overview of the experiments). Gas-phase instruments included a quadrupole proton



transfer reaction mass spectrometer (Q-PTR-MS, Ionicon), an ozone monitor (Thermo 49C), and a NO$_x$ monitor (Thermo 42C). Particle-phase instruments included a scanning mobility particle sizer (SMPS, TSI model 3938), a custom-built VTDMA, isothermal evaporation chambers coupled to a custom-built SMPS, and a FIGAERO-CIMS (Aerodyne). Clean air was supplied by a zero air generator (AADCO), and was used to flush the chamber overnight following every experiment at

50 L min$^{-1}$ with the chamber volume heated to 30°C.

In the NO$_3$ experiments, the chamber was humidified with milliQ water (resistivity > 18 MΩ cm) until it reached the desired humidity, then the VOC of choice (α-pinene – Sigma Aldrich 99+%, isoprene – Sigma Aldrich 95+%, or β-caryophyllene – Sigma Aldrich 99+%) was injected into the chamber, and finally N$_2$O$_5$ (sampling from the head space above solid N$_2$O$_5$, which was kept chilled in an ice bath) was injected in a short burst (~5 seconds at ~1 L min$^{-1}$) and the chamber was

stirred by injecting 100 L min$^{-1}$ of air for 10 seconds. N$_2$O$_5$ was generated by reacting liquid NO$_2$ with high concentrations (~1%) of O$_3$ and collecting the N$_2$O$_5$ crystals in a dry ice/acetone bath. In the O$_3$ experiments, the chamber was first humidified, followed by O$_3$ addition, produced by photolysis of air, to the chamber to reach the desired levels of ~340 ppb, then α-pinene was added to the chamber volumetrically.

To determine the initial N$_2$O$_5$ concentrations, the decay rates of the VOC precursors were modelled with the Master

Chemical Mechanism (MCM) using a simple 0-D box model (F0AM, Wolfe et al., 2016). For the β-caryophyllene experiment, its concentration could not be monitored with the PTR-MS because its mass-to-charge ratio (*m/z*) of 204 Th is outside of the mass transmission range for quantitative measurements, thus we estimated the concentration from the average N$_2$O$_5$ concentrations in the other experiments.

The particle mass concentrations (in μg m$^{-3}$) were derived from integrated number size distributions from the SMPS

and their conversions to mass using an assumed organic aerosol density (1.19 g cm$^{-3}$; Vaden et al., 2011). They were further corrected for wall losses using a uniform dynamic wall loss rate $k_{wall}$ for the whole size range. The wall loss rate ($k_{wall}$) was calculated from the exponential decay of the total particle number concentration (cm$^{-3}$) measured by the SMPS, corrected for coagulation. The particulate wall loss is defined by:

$$\frac{dN}{dt} = -k_{coag.}N^2 - k_{wall}N \qquad\qquad\qquad \text{(Eq. 1)}$$

where $N$ is the particle number concentration and $k_{coag.}$ corresponds to the coagulation coefficient (5 × 10$^{-10}$ s$^{-1}$, Pospisilova et al., 2020). The wall loss-corrected particle mass was divided by the uncorrected mass to obtain a wall loss correction factor for correction of the signals from the FIGAERO-CIMS.



**Table 1. Overview of the experimental conditions, maximal OA load and SOA yield together with initial particle diameter ($D_{sel.}$) and concentration ($C_{sel.}$) used as input in the kinetic model for the thermodenuder and isothermal evaporation. All parameters are derived from the experimental data.**

| Reaction | Experiment plot symbol[a] | | Reacted VOC (ppb) | $O_3/N_2O_5$ (ppb) | Max. OA load ($\mu g\ cm^{-3}$) | SOA yield (%) | Thermodenuder | | Isothermal evap. | |
|---|---|---|---|---|---|---|---|---|---|---|
| | | | | | | | $D_{sel.}$ (nm) | $C_{sel.}$ ($cm^{-3}$) | $D_{sel.}$ (nm) | $C_{sel.}$ ($cm^{-3}$) |
| **α-Pinene + O₃** | | | | | | | | | | |
| | 1 | ▲ | 50 | 330 | 47 | 17 | - | - | 133 | 40420 |
| | 2 | ▼ | 40 | 350 | 23 | 10 | 52 | 2610 | - | - |
| **α-Pinene + NO₃** | | | | | | | | | | |
| | 3 | | 20 | 400 | 8 | 7 | - | - | - | - |
| | 4 | | 60 | - | 30 | 9 | - | - | - | - |
| | 5 | ▲ | 100 | 300 | 18 | 3 | - | - | 133 | 40420 |
| | 6 | ● | 100 | 80 | 39 | 7 | 110 | 2314 | 140 | 46680 |
| | 7 | ▼ | 100 | 300 | 53 | 10 | 242 | 123 | 137 | 24186 |
| **Isoprene + NO₃** | | | | | | | | | | |
| | 8 | ▲ | 100 | 200 | 9 | 3 | 104 | 2271 | - | - |
| | 9 | ● | 100 | 120 | 24 | 9 | 115 | 1884 | 151 | 39720 |
| | 10 | ▼ | 100 | 180 | 11 | 4 | 87 | 2784 | 116 | 13758 |
| **β-Caryophyllene + NO₃** | | | | | | | | | | |
| | 11 | ▲ | 50 | >200 | 464 | 111 | 104 | 4110 | 100 | 211371 |
| | 12 | ▼ | 5 | >200 | 60 | 144 | 104 | 1444 | 110 | 47396 |

[a] Symbols used in Figs. 4, 5 and S5 to mark each experiment.

## 2.2 Volatility measurements

We assessed the SOA volatility using three methods: 1) a VTDMA (2.2.1); 2) an isothermal evaporation chamber (2.2.2); and 3) a FIGAERO-CIMS (2.2.4). From the first two methods, we observed reduced particle diameters due to heating and dilution, respectively, and calculated the volume fraction remaining (VFR) (see 2.2.3). With the FIGAERO-CIMS, we obtained thermal desorption profiles of individual compounds as well as the bulk sample. In addition, we also estimated the volatility based on the chemical composition measured by FIGAERO-CIMS, and modelled the temperature- and time-dependent VFR using a kinetic evaporation model (2.2.5) for comparison with the VTDMA and the isothermal evaporation chamber, respectively.

### 2.2.1 Volatility tandem differential mobility analyzer (VTDMA)

A custom-built VTDMA was used to measure the change in particle diameter upon heating. The VTDMA (see e.g. Tritscher et al., 2011) sampled aerosol particles through a silica gel diffusion drier connected to an approximately 5.5 m long stainless steel tube reaching to the chamber. The VTDMA combines two differential mobility particle sizer (DMPS) systems, coupled in series with a thermodenuder (TD) in between. Both DMPS systems consisted of a custom-made short differential mobility analyzer (DMA, Stockholm University, operated with closed-loop sheath air and 1 L/min sample flow) coupled to a condensation particle counter (CPC, TSI 3010). Before entering each DMA system, the particles were brought into charge equilibrium using a Ni-63 neutralizer.

The first DMA selected a nearly monodisperse aerosol with the diameter ($D_{sel.}$) set to the geometric mean diameter of the aerosol population (varying from 52 to 242 nm between experiments) measured by the SMPS upstream. The sample





flow was split in two, with half going to the CPC measuring the selected particle concentration ($C_{sel.}$) and the other half entering a 35 cm long custom-built TD, with a residence time of 1.9 s. For each experiment a set of temperature ramps from 50 to 225 °C in increments of 25 °C were performed with the TD. The size distribution of the heated aerosol was measured in the second DMPS system with a time resolution of approximately 5 min.

### 2.2.2 Isothermal evaporation chamber

Isothermal evaporation measurements were performed approximately 1-5 h into the experiment. Nearly monodisperse aerosol samples was size-selected with a DMA (TSI Inc. Model 3082) and placed into a 20 L stainless steel chamber, which operated at room temperature (20 °C, set by the air conditioning in the instrument container). The chamber was equipped with a layer of charcoal at the bottom of the chamber. The charcoal continuously removed the gas phase species from the chamber ensuring the particles never fully reached equilibrium with the surrounding gases, promoting evaporation. Changes in particle sizes due to isothermal evaporation were monitored using an SMPS (TSI Inc., model 3938). The experimental setup has been described in detail elsewhere (Bell et al., 2017; Vaden et al., 2011; Wilson et al., 2015).

### 2.2.3 Volume fraction remaining (VFR)

As particles evaporate in the VTDMA and isothermal evaporation chamber, they shrink in size, and the resulting volume fraction remaining (VFR) can be used as proxy for the aerosol population volatility. We calculated the VFR values for the VTDMA and evaporation chamber experiments from the particle diameters as follows:

$$\text{VFR} = \frac{D_{evap.}^3}{D_{sel.}^3} \qquad (\text{Eq. 2})$$

where $D_{sel.}$ is the initial geometric mean mobility particle diameter and $D_{evap.}$ is the size observed after evaporation took place. In the VTDMA, the $D_{evap.}$ values are the heated diameters, represented by the geometric mean diameter estimated using a Gaussian fit over the whole heated size distribution. The uncertainty in the temperature-dependent VFR is the sum of the standard deviation of the VFR and the error of the two DMPS systems within the VTDMA, estimated using error propagation law and polystyrene latex (PSL) spheres for calibration. For the evaporation chamber experiments, $D_{evap.}$ was directly reported by the SMPS coupled to the chamber.

### 2.2.4 Deriving the volatility distribution from the FIGAERO CIMS data

The molecular composition of the generated SOA was measured using the FIGAERO-CIMS, whose design, operation and related data analysis are described in C. Wu et al. (2021). Briefly, the FIGAERO-CIMS was set up using iodide (I⁻) as reagent ion and an X-ray generator as ion source. Particles were collected on a 25 mm Zefluor® PTFE filter (Pall Corp.) with 2 µm pore size via a sampling port directly connected to the chamber (flow rate 5 L/min). The duration of sampling was 10 – 20 min for most of the experiments. When the sampling was done, particles on the filter were desorbed by a flow of gradually heated ultra-high-purity (99.999 %) nitrogen. A FIGAERO-CIMS desorption round lasted 40 min: 20 min of ramping the temperature




of the nitrogen flow up to 200 °C, followed by a 20 min "soak period" at a constant 200 °C to allow signal to go back to background values. After that it was cooled down to room temperature (for 15 min). Artefacts from thermal fragmentation (< 30% of total signal for all experiments, details see Wu et al., 2021) were subtracted for further analysis. The potential uncertainties induced by thermal fragmentation are discussed in Section 3.4.

For each individual filter sample and the ensemble of deposited particles, we estimated the volatility of individual
compounds based on their molecular composition measured by the FIGAERO-CIMS, and further binned their concentrations into a volatility distribution. The effective saturation vapor concentration ($C^*$) of individual compounds was calculated using three different previously published parameterizations which have explicit formulations for nitrate groups/nitrogen:

1) an updated version of the parameterization by Donahue et al. (2011), modified based on the saturation vapor concentrations of highly oxygenated molecules (HOMs) detected by Tröstl et al. (2016) and published by Mohr et al. (2019):

$$\log_{10}C^* = (n_0 - n_C)b_C - (n_O - 3n_N)b_O - 2\frac{n_C(n_O - 3n_N)}{n_C + n_O - 3n_N}b_{CO} - n_N b_N \qquad \text{(Eq. 3)}$$

where $n_0 = 25$, $b_C = 0.475$, $b_O = 0.2$, $b_{CO} = 0.9$ and $b_N = 2.5$. $n_C$, $n_O$ and $n_N$ are the number of carbon, oxygen and nitrogen atoms in the compound, respectively. This parameterization hence assumes that a nitrate group (1 nitrogen atom and 3 oxygen atoms) reduces a compound´s saturation vapor pressure by 2.5 orders of magnitude (Donahue et al., 2011; Pankow and Asher, 2008).

2) an updated version of Li et al. (2016) with a modified nitrogen coefficient for organic nitrates (Isaacman-VanWertz
and Aumont, 2020):

$$\log_{10}C^* = (n_0 - n_C)b_C - (n_O - 3n_N)b_O - 2\frac{n_C n_O}{n_C + n_O}b_{CO} - n_N b_N \qquad \text{(Eq. 4)}$$

where $n_0 = 22.66$, $b_C = 0.4481$, $b_O = 1.656$, $b_{CO} = -0.7790$ for CHO compounds, and $n_0 = 24.13$, $b_C = 0.3667$, $b_O = 0.7732$, $b_{CO} = -0.07790$ and $b_N = 0.7732$ for CHON compounds. The modification of the parameterization is based on the observation that a nitrate group has a similar impact on the saturation vapor pressure as a hydroxyl group, thus reduces a compound's vapor
pressure by 0.7732 orders of magnitude ($b_N = b_O$, (Isaacman-VanWertz and Aumont, 2020).

3) a simple parameterization based on highly oxygenated organic molecules (HOMs) from α-pinene ozonolysis (Peräkylä et al.,2020):

$$\log_{10}C^* = 0.18 \times n_C - 0.14 \times n_H - 0.38 \times n_O + 0.80 \times n_N + 3.1 \qquad \text{(Eq. 5)}$$

where $n_H$ is the number of hydrogen atoms in the compound. With this parameterization, a nitrate group reduces a compound's
saturation vapor pressure by 0.34 orders of magnitude.

The temperature dependence of $C^*$ is considered as follows:

$$C^*(T) = C^*(300 \text{ K}) \exp\left(\frac{\Delta H^{\text{VAP}}}{R}\left(\frac{1}{300 \text{ K}} - \frac{1}{T}\right)\right) \qquad \text{(Eq. 6)}$$

where $T$ is the temperature in Kelvin, $C^*(300 \text{ K})$ is the saturation vapor concentration at 300 K, $R$ is the gas constant, and $\Delta H^{\text{VAP}}$ is the vaporization enthalpy (Epstein et al., 2010):

$$\Delta H^{\text{VAP}} = -11\log_{10}C^*(300 \text{ K}) + 129; \qquad \Delta H^{\text{VAP}} < 200 \text{ kJ/mol} \qquad \text{(Eq. 7)}$$





Qualitative volatility information was also derived from the FIGAERO-CIMS desorption profiles referred to as thermograms. We calculated the mass-weighted average thermogram and corresponding $T_{max}$ of the entire particle population deposited on the filters from the FIGAERO-CIMS. The desorption temperature at which a compound´s signal exhibits a maximum in the FIGAERO-CIMS (referred to as $T_{max}$), has earlier been shown to be connected qualitatively to the volatility

of the compound ($\log_{10}C^*$ or enthalpy of vaporization) (Bannan et al., 2019; Lopez-Hilfiker et al., 2014; Thornton et al., 2020).

### 2.2.5 Kinetic evaporation modelling

We used a kinetic evaporation model (Riipinen et al., 2010) to predict the evaporation of the aerosol populations in the TD and the isothermal evaporation chamber, i.e. temperature- and time-dependent dependent VFR, respectively. The model solves the equations describing the mass transfer between the particulate and the gas phase due to condensation and evaporation. The

model is applicable over a wide range of particle sizes, as it describes the transport over kinetic, transition and continuum regimes using the Fuchs and Sutugin transition regime correction factor (see Riipinen et al., 2010 and the references therein for details). The particle size distribution and volatility distribution, initial gas-phase concentrations of the relevant species and the temperature evolution are given as inputs to the model – together with the molecular properties of the chemical components of the system (see Table S1). The initial particle diameter was set to the same sizes as the monodisperse size selection prior

each TD and isothermal evaporation measurement (see Table 1). Additionally, the initial particle concentration for each $D_{sel.}$ was set to the corresponding concentration measured by the CPC within the VTDMA and using the SMPS for the TD and isothermal evaporation chamber, respectively (see Table 1). For the VTDMA setup, an initial temperature corresponding to room temperature (20 °C), and a flat temperature profile (constant temperature throughout the TD) with final temperatures varying between 25 and 225 °C together with a residence time of 1.9 s were used for the TD section. The gas phase was

assumed to be initially in equilibrium with the particle phase for the VTDMA setup. For the isothermal evaporation chamber setup, the gas phase was assumed to be effectively stripped from the condensable vapors throughout the evaporation, and the evaporation took place at room temperature (20 °C) over the residence time of 240 s.

In addition to the physical and chemical properties presented in Table S1, the model input included saturation vapor concentrations estimated based on the chemical composition measured by the FIGAERO-CIMS. The three volatility

parameterizations (see previous section) were used as model input for the $C^*$ values of the SOA compounds, and the corresponding $\Delta H^{VAP}$ were calculated using Eq. 7. The sensitivity of the model setup was evaluated (see Fig. S3) also for a fixed vaporization enthalpy, $\Delta H^{VAP} = 70$ kJ/mol (see e.g. Hong et al., 2017), reduced accommodation ($\alpha_m$, set to be either 0.1 or 0.01, see e.g. Riipinen et al., 2010), and a mass-dependent diffusion coefficient, estimated by the method by Fuller et al. (1965, 1966, 1969, see Poling et al. (2001) for more details) from the measured compositions:

$$D_{AB} = \frac{0.00143\,T^{1.75}}{P\,M_{AB}^{1/2}\left[(\Sigma_v)_A^{1/3}+(\Sigma_v)_B^{1/3}\right]^2}\;; \qquad M_{AB} = 2[(1/M_A) + (1/M_B)]^{-1} \qquad \text{(Eq. 8)}$$

where $T$ is the temperature (K), $P$ is the pressure (1 bar), $M$ is the molecular weight (g mol$^{-1}$) for gases A and B, the latter being air and the former the organic vapor. $\Sigma_v$ is the sum of the atomic diffusion volumes of the two components (see





Table S1). Particle-phase diffusion or any chemical reactions during the evaporation were ignored as a first-order approximation.

## 3 Results and discussion

### 3.1 SOA formation and yields

An overview of the initial precursor and oxidant concentrations is presented in Table 1, together with the resulting maximal mass load in the particle phase. Prompt particle formation, due to rapid reaction with $NO_3$ or $O_3$, followed by steady particle mass decay due to evaporation, was observed for all three precursors. For the $NO_3$ experiments, the initial particle composition followed primarily from the reactions between peroxy radicals ($RO_2$-$RO_2$) and between $RO_2$ and $NO_3$, resulting in a substantial fraction of oligomers with the dimer-to-monomer ratio increasing with decreasing precursor carbon number (Bates et al., 2021; Bell et al., 2021; C. Wu et al., 2021). According to the FIGAERO-CIMS, the mass fractions of oligomers were 86–88 %, ~100 % and 60–63% for the α-pinene, isoprene and β-caryophyllene SOA, respectively. Compared to the nitrate systems, the oligomer mass fraction for α-pinene + $O_3$ was lower and in the range of 45–60 %. The aerosol particles formed from nitrate oxidation of α-pinene and isoprene were dominated by oligomers with two and 3-4 nitrate groups, respectively. The β-caryophyllene SOA was dominated by both monomers with 1-2 nitrate groups and dimers with 2-3 nitrate groups (C. Wu et al., 2021). The wall-loss-corrected maximum mass concentrations formed were 23–47 µg m$^{-3}$ for α-pinene + $O_3$, 8–53 µg m$^{-3}$ for α-pinene + $NO_3$, 9–24 µg m$^{-3}$ for isoprene + $NO_3$, and 60–464 µg m$^{-3}$ for β-caryophyllene + $NO_3$. The corresponding SOA yields of 10-17 % (α-pinene + $O_3$) , 3–10 % (α-pinene + $NO_3$), 3–9 % (isoprene + $NO_3$) and 111–144 % (β-caryophyllene + $NO_3$) are generally in line with the literature (Berkemeier et al., 2020; Ng et al., 2017; R. Wu et al., 2021).

### 3.2 Measured evaporation of SOA from α-pinene, isoprene and β-caryophyllene oxidation

Figure 1 presents the experimental data on the evaporation of α-pinene SOA from oxidation with $O_3$ and $NO_3$, measured as temperature-dependent VFR in the VTDMA (Fig. 1a), time-dependent VFR in the isothermal evaporation chamber (Fig. 1b) and temperature-dependent normalized signal (thermograms) during heating rounds from the FIGAERO-CIMS (Fig. 1c). Comparison of the results with the two oxidants in Fig. 1a reveals that particles formed through nitrate oxidation of α-pinene consistently display a lower VFR at different temperatures from 25 to 225°C and hence higher bulk volatility than the particles from ozonolysis under our experimental conditions for the VTDMA. We also compared the VFR data from the α-pinene ozonolysis experiments to available literature data (see e.g. Tritscher et al., 2011, and references therein) in Fig. S1. The comparison shows general agreement with the previous measurements, with some variation arising from the different experimental conditions such as the residence time within the TD, which ranges from < 1 s up to 31.6 s in the data sets included in Fig. S1. With the residence time of 1.9 s, our measurements are at the short end of the range, with correspondingly higher VFR to be expected. The lower VFRs during isothermal evaporation for $NO_3$ oxidation as compared with ozonolysis agree the VTDMA data, indicating that nitrate oxidation of α-pinene SOA produces more volatile compounds (Fig. 1b). The FIGAERO-





CIMS average thermogram also reveals lower maximum desorption temperature, indicating higher bulk volatility for α-pinene

+ NO₃ than for α-pinene ozonolysis (Fig. 1c). The three methods are therefore overall consistent in this regard. The time-dependent VFR for nitrate SOA with different loadings in the isothermal evaporation exhibits a tendency for lower VFR for higher initial mass loadings. This may be related to e.g. variation in the volatility distributions depending on the precursor and oxidant loadings. However, no significant dependence of the evaporation on the mass loading was observed for the VTDMA results (see Fig. S2).

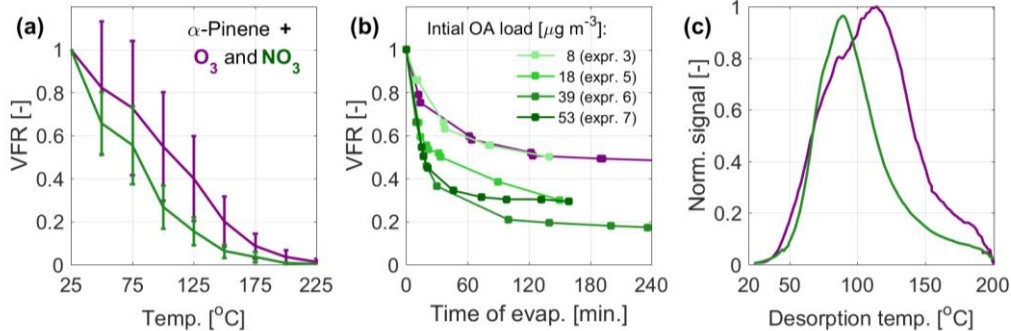


**Figure 1. Measured volatility of SOA particles from α-pinene oxidation by O₃ (purple) and NO₃ (green). a) Mean VFR evolution during the VTDMA temperature ramps; b) the isothermal evaporation; c) mean thermograms measured by the FIGAERO-CIMS. The VFR at room temperature (a) is estimated based on predicted evaporation at room temperature (b). For the VTDMA experiments, the VFR values correspond to the mean of two consecutive recordings during one ozonolysis experiment and the mean**

**of the four individual nitrate oxidation experiments (see Table 1), where the uncertainty bars correspond to the sum of the standard deviation of the VFR and the error of the two DMPS systems within the VTDMA (see Sect. 2.2.3).**

Figure 2 presents the volatility data from the VTDMA, isothermal evaporation chamber and FIGAERO-CIMS for the nitrate oxidation of isoprene (upper row, Figs. 2a-c) and β-caryophyllene (lower row, Figs. 2d-f) in comparison with α-pinene. The isoprene + NO₃ SOA has similar bulk volatility as the α-pinene + NO₃ SOA. Both the temperature ramp in the VTDMA and

residence time in the isothermal evaporation chamber result in slightly higher VFR and hence somewhat lower volatility than for the corresponding SOA from α-pinene oxidation. The effect is not as clear in the FIGAERO-CIMS thermogram, where the peak of the bulk thermogram is observed at slightly lower temperatures than for α-pinene, although with a prominent shoulder at higher temperatures. The temperature-dependent VFRs for the β-caryophyllene system are even higher, and in the isothermal chamber no significant evaporation could be observed. In the FIGAERO-CIMS analysis, the β-caryophyllene data display a

bimodal thermogram with one peak (from monomers) at the same temperature as α-pinene, while the dominant peak (from dimers) is located at higher temperatures, in agreement with the higher measured VFRs.



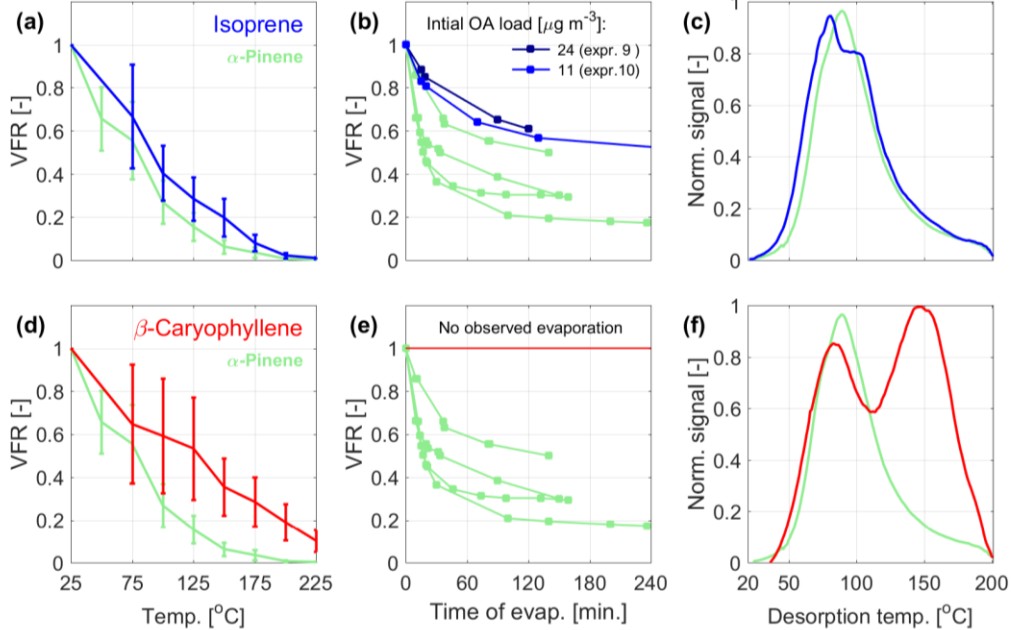

**Figure 2 Measured volatility of SOA from isoprene (a-c, blue curves) and β-caryophyllene (d-f, red curves) oxidation by NO₃. The corresponding data from α-pinene oxidation are marked in pale green in the background. The left column (a, d) presents the mean**
**VFR evolution during the VTDMA temperature ramps, the middle column (b, e) the isothermal evaporation, and the right column (c, f) mean thermograms measured by the FIGAERO-CIMS. The VFR at room temperature (a, d) is estimated based on predicted evaporation at room temperature (b, e). For the VTDMA experiments, the VFR values correspond to the mean of three and two individual experiments for isoprene and β-caryophyllene, respectively (see Table 1), where the uncertainty bars correspond to the sum of the standard deviation of the VFR and the error of the two DMPS systems within the VTDMA (see Sect. 2.2.3).**

### 3.3 Volatility distributions derived from the molecular composition

The volatility distributions in the form of VBS resulting from the parameterizations by Mohr et al. (2019), Isaacman-VanWertz and Aumont (2021) and Peräkylä et al. (2020) (hereafter referred to as MHR, IvWA and PRK, respectively) are presented in Fig. 3 for all the studied chemical systems, including α-pinene ozonolysis. PRK produces the highest volatilities, followed by IvWA, with MHR resulting in the least volatile compounds. The volatility distributions from MHR and IvWA tend to contain

a wider range of $C^*$ values, while the volatility distributions by PRK are narrower and generally unimodal; the same trend can be observed for all the four systems.




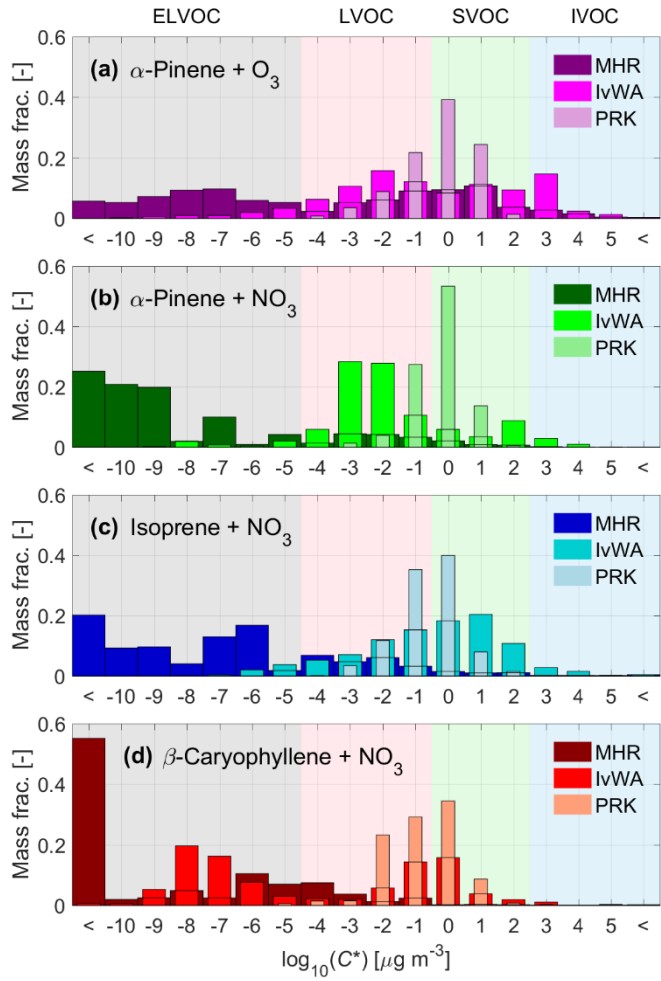

**Figure 3. Volatility distributions in the form of VBS as derived from the molecular composition for α-pinene ozonolysis (a) and nitrate oxidation of α-pinene (b), isoprene (c) and β-caryophyllene (d). The three volatility parameterizations used (IvWA: Isaacman-VanWertz and Aumont 2021; MHR: Mohr et al., 2019; PRK: Peräkylä et al., 2020) are depicted with different shades of the main color for each of the precursors.**

The MHR parameterization predicts the presence of extremely low volatilities for a major part of the nitrate oxidation products of all three different precursors, with β-caryophyllene (Fig. 3d) producing the least volatile compounds, which is qualitatively in line with the observed VFRs (see Fig. 2). For α-pinene and isoprene, MHR predicts the majority of the compounds to fall into the volatility bins located below $\log_{10}(C^*) \sim -6$ and -5, respectively. For β-caryophyllene, MHR predicts two main classes of volatilities: one below $\log_{10}(C^*) \sim -10$ (oligomers) and another, broader peak around $\log_{10}(C^*) \sim -5.5$ (monomers). Comparing the nitrate system with the ozonolysis system, MHR predicts that α-pinene nitrate (Fig. 3b) SOA has lower volatility than α-pinene ozonolysis SOA(Fig. 3a). This result is in contrast with the direct volatility measurements displaying



lower VFR, hence higher volatility for nitrate oxidation compared to α-pinene ozonolysis (see Fig. 1). All in all, the MHR
parameterization predicts ELVOCs to dominate the composition of the SOA from nitrate oxidation of BVOCs.

The other two parameterizations yield narrower distributions that are generally shifted towards higher volatility. PRK
results in higher volatilities than IvWA for all three precursors, with the majority of the species being in the LVOC and SVOC
range. With the IvWA parameterization, the isoprene nitrate products are more volatile than the α-pinene nitrate products, with
isoprene peaking around $\log_{10}(C^*) \sim 0.5$ compared to $\log_{10}(C^*) \sim -2.5$ for α-pinene. For the SOA from β-caryophyllene, IvWA
also produces a bimodal distribution, overlapping both the MHR and PRK distributions. The least volatile peak for IvWA
overlaps the most volatile peak in the MHR distribution, at $\log_{10}(C^*) \sim -7.5$, while the more volatile peak appears at $\log_{10}(C^*)$
$\sim -0.5$, instead overlapping with the whole distribution by PRK for β-caryophyllene. Finally, for the PRK parametrization,
there is no big difference in the peak of the volatility distribution between the different precursors, suggesting this
parameterization to be the least sensitive to the details of the molecular composition. Although the differences in the VBS for
the α-pinene nitrate- and ozonolysis system are smaller using these two parameterizations, they still predict lower volatility
products of the nitrate system compared to the α-pinene ozonolysis SOA, which is in contrast with the observations. Clearly,
the three parameterizations yield different volatility distributions, which are not always in line with the measured evaporation
described in Section 3.2.

## 3.4 Comparison to evaporation calculated with a kinetic model

In order to better compare the volatility derived from the chemical composition to the evaporation observed in the VTDMA
and isothermal evaporation chamber, the volatility distributions derived from the three investigated parameterizations (Sect.
3.3, Fig. 3) were entered into a kinetic model describing the mass transport to/from the aerosol surface (Riipinen et al., 2010,
Sect. 2.2.5). The model results are compared to the experiments in Fig. 4.

The calculated evaporation of SOA from α-pinene ozonolysis (Fig. 4, first column) using MHR reproduces the
evaporation well, while the other two parameterizations, IvWA and PRK, result in lower VFR compared to the measurements.
The overall performance is consistent for both VTDMA and isothermal evaporation data. The α-pinene ozonolysis dataset was
used to demonstrate the sensitivity of the model to the key input parameters (see Table S1). In this regard we tested the values
of 0.1 and 0.01 for the accommodation coefficient (see e.g. Riipinen et al., 2010), a fixed value of 70 kJ/mol for the vaporization
enthalpy (see e.g. Hong et al., 2017) as well as the effect of estimating the diffusion coefficient based on the molecular
composition, which was now directly available from the FIGAERO-CIMS data (see Eq. 8). The results are presented in Fig
S3. While modifying the accommodation coefficient or the vaporization enthalpy can bring the other parameterizations
somewhat closer to the observations, it is clear that the MHR parameterization captures the evaporation of this system
exceptionally well with the base case assumptions in the model. Including a detailed description of the diffusion coefficient
based on the molecular structure had no significant effect on the output of the data for any of the volatility parameterizations.






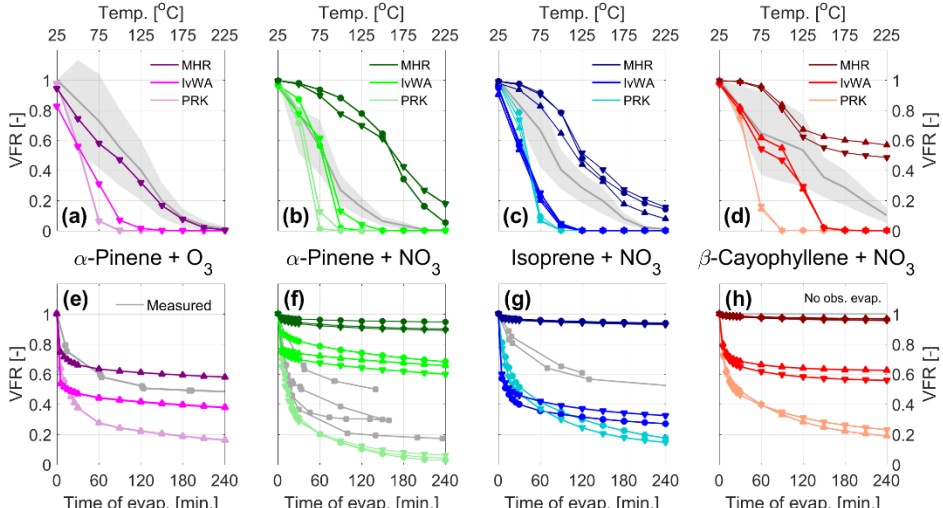

**Figure 4. Evaporation in the TD (upper row) and isothermal evaporation chamber (lower row) predicted by the kinetic model using the VBS from Fig. 3 as input for α-pinene ozonolysis (a and e, purple shades) and nitrate oxidation of α-pinene (b and f, green shades), isoprene (c and g, blue shades) and β-caryophyllene (d and h, red shades). The symbols (▲●▼) mark the experiments in chronological order, as indicated in Table 1, during which the VBS data was collected. For comparison, the experimental VFR data are displayed in the background in grey, with the shaded areas displaying the uncertainty of the VTMDA measurement.**

In contrast to the excellent performance for the α-pinene ozonolysis, the MHR parameterization results in too high VFR for the nitrate oxidation of α-pinene (Fig. 4, second column) compared to the measurements, and higher VFR for nitrate oxidation compared to α-pinene ozonolysis. The other two parameterizations result in lower VFR than MHR, with especially IvWA performing better for the α-pinene nitrate oxidation. These results thus suggest that the extremely low volatilities predicted by MHR for the organic nitrates are not realistic (see Fig. 3). Similarly, for the nitrate oxidation of isoprene, the parameterization from MHR results in over-predicted VFR in both the VTDMA and the isothermal evaporation setups (Fig. 4, third column), while IvWA and PRK under-predict the VFR. For the β-caryophyllene case (Fig. 4, fourth column), predictions based on IvWA and PRK also underestimate VFR. MHR reproduces best the lack of observed evaporation in the isothermal evaporation chamber, but overestimates the VFR in the VTDMA setup for β-caryophyllene. Based on these results, no parameterization can be deemed to be systematically the best for reproducing both the VTDMA and the isothermal evaporation chamber data. This is understandable, since these experimental techniques are sensitive to different volatility ranges given the different operating temperatures. Combining these results with the volatility distributions (Fig. 4) and the results shown in Fig. S4 suggests that the material contributing to the VFRs from the VTDMA setup consist of species with volatilities at the low end of the spectrum ($\log_{10}(C^*)$ values of about -5 and lower), while the species that remain in the particle phase in the isothermal evaporation chamber seem to have $\log_{10}(C^*)$ values below about -2. This is especially clear for the isoprene simulations, with both the TD and isothermal evaporation simulations being very sensitive in the volatility range between $\log_{10}(C^*) \sim$ -5 to -2 (see Figs. 3 and 4).

It is also worth noting that with the FIGAERO-CIMS, subtraction of the thermal decomposition compounds may also induce uncertainties. Thermal fragmentation contributed 5%–27%, 1%–4%, and 10%–23% of the total organic signal of the





isoprene, α-pinene, and β-caryophyllene nitrate SOA, respectively, and 3–13% for α-pinene ozonolysis SOA. Compared to the measured thermal fragmentation products, their parent compounds have larger masses and also higher thermal desorption temperature, thus lower volatility. Though proper identification is not possible with our method, their contribution to the ELVOC and LVOC range should not be ignored. It could be a potential reason for the discrepancies between the calculated

VFR and the measured VFR, especially with the VTDMA, as the VTDMA has a similar heating process. In addition, as α-pinene nitrate SOA exhibits much less thermal fragmentation compared to isoprene and β-caryophyllene, it could also explain partly the different performance of parameterization in different systems.

Taken together, the results from Figs. 4 and 5 suggest that the volatility reduction by the nitrate group is considerably overestimated by the MHR parameterization, while the other two parameterizations tend to systematically overestimate the

volatility of the studied systems in the TD and isothermal evaporation setups. This is generally in line with previous findings, particularly the conclusion by R. Wu et al. (2021) for the isoprene + NO$_3$ system, who find that the MHR parameterization gives clearly lower volatilities than other parameterizations and experimental approach developed by the authors. Furthermore, these results show a significant contribution of species in the LVOC range in the SOA from both α-pinene + NO$_3$ and isoprene + NO$_3$, and probably the presence of SVOCs, which show significant evaporation even at room temperature (since the

isothermal evaporation is mainly sensitive in this range, as described above). The SOA from β-caryophyllene + NO$_3$, on the other hand, seems to consist primarily of species in the ELVOC and LVOC range. Our data presents overall lower volatilities of SOA formed from α-pinene + NO$_3$ and isoprene + NO$_3$ compared with the few other existing studies (Berkemeier et al., 2020; Boyd et al., 2015; Ng et al., 2017; R. Wu et al., 2021), while for the β-caryophyllene + NO$_3$ system no previous data exists. When interpreting these comparisons, however, one should be aware of the somewhat different experimental conditions

in the existing studies, remembering e.g. the fact that our experiments show a significant contribution from oligomers.

The MHR and IvWA parameterizations are of similar mathematical form, with the exception that the former subtracts the oxygen atoms from the nitrate groups in the carbon-oxygen non-ideality, while the latter includes all oxygen atoms in the carbon-oxygen non-ideality. In MHR, a nitrate group reduces the saturation vapor pressure of a compound by 2.5 orders of magnitude, instead of 0.7732 orders of magnitude in IvWA (which is similar as a hydroxyl group). The difference is due to

the different training datasets used to create the parameterizations. The PRK parameterization, on the other hand, is based on experimental methods, specifically on HOMs from α-pinene ozonolysis, and displays a simple dependence on the individual atoms instead of specific functional groups (like the nitrate group). With this in mind it is not surprising that PRK best captures α-pinene. Finally, the uncertainty of the effect of nitrogen groups on the saturation vapor pressures of organic species is also to be expected, given the lack of quantitative experimental data to use to train the parameterizations (see e.g. Pankow and

Asher, 2008). Therefore, we investigated the effect of modifying the parameter $b_N$ in steps of 0.25 for MHR and IvWA for better reproducing the observations (see Table 2 and Fig. 5). When applying the MHR parameterization, the nitrate-induced reduction in saturation vapor pressure needed to be substantially reduced for all three systems to fit the predicted evaporation to the observations. For the isoprene system, $b_N$ values between 1.0 and 1.5 provided the best fit to the observations (Fig. 5b), still however indicating a saturation vapor pressure reducing effect of the nitrate group. For the α-pinene and β-caryophyllene





systems, however, the $b_N$ values needed to be reduced to negative values (from –0.75 to –0.5) to fit the observations (Figs. 6a, c). The negative values would indicate that the nitrate group would increase the volatility.

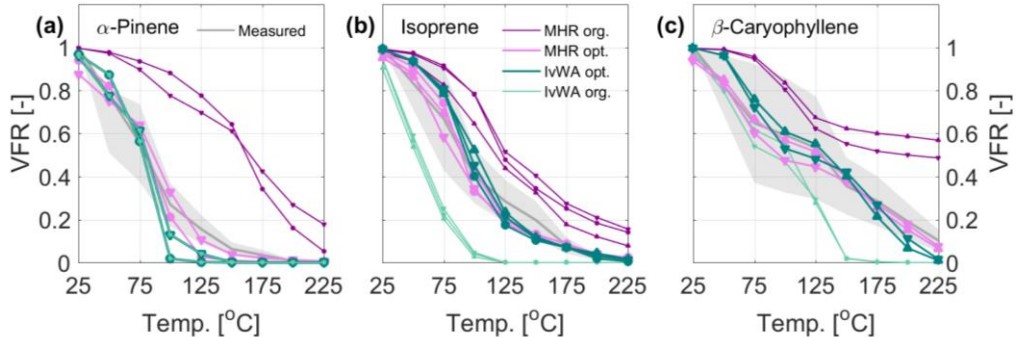

**Figure 5. Evaporation in the TD predicted by the kinetic model after optimizing the nitrate influence in MHR (pink) and IvWA (mint) for the nitrate oxidation of (a) α-pinene, (b) isoprene and (b) β-caryophyllene. The symbols (▲●▼) mark the experiments in**
**chronological order, as indicated in Table 1, during which the VBS data was collected. For comparison, the original modeled (MHR in purple and IvWA in teal) and experimental evaporation (in the background in grey) are displayed.**

In the case of IvWA, the original value of $b_N = 0.7732$ already performed exceptionally well for the nitrate oxidation of α-pinene, and hence no adjustment was warranted here (Fig. 5a). For isoprene and β-caryophyllene, the saturation vapor pressure reduction by the nitrate group needed to be increased to 1.8 – 2.8 orders of magnitude, indicating similar nitrate influence as
previously estimated by Pankow and Asher (2008). Given that no $b_N$ could reproduce the data from the different precursors for neither MHR nor IvWA, it is obvious that only adjusting the $b_N$ value could not fix the discrepancy we observed between the measured and the simulated VFR based on the chemical composition. The $b_N$ value is closely related to the other interaction terms, e.g. the carbon-carbon interaction term and the carbon-oxygen nonideality. This finding indicates complexity of the interactions between different functional groups and warrants further and more detailed studies on nitrate-containing systems.

**Table 2. Optimal nitrate influence ($b_N$) for MHR and IvWA tuned using α-pinene, isoprene and β-caryophyllene.**

|  | Parameterization | α-Pinene | | Isoprene | | | β-Caryophyllene | |
|---|---|---|---|---|---|---|---|---|
| Expr. nr | | 6 | 7 | 8 | 9 | 10 | 11 | 12 |
| $b_N$ | MHR | -0.75 | -0.5 | 1.5 | 1.25 | 1.0 | -0.5 | -0.5 |
| | IvWA | 0.7732 | 0.7732 | 2.8 | 2.15 | 2.25 | 1.8 | 1.9 |



**Conclusions**

Here we studied the volatility of SOA particles from nitrate oxidation of α-pinene, isoprene and β-caryophyllene using a set
of experimental techniques, presenting a cohesive picture of the three studied precursors. The experimental data from this work
as well as the comparison to current literature reveal that under our experimental conditions, the SOA formed from nitrate
oxidation of α-pinene is more volatile than the corresponding α-pinene ozonolysis products. Further, the nitrate oxidation of
β-caryophyllene produces less volatile SOA than α-pinene and isoprene.

Besides observing the volatility directly through following the evaporation of the particles, the particle-phase
composition observed by the FIGAERO-CIMS was used to predict the volatilities present in the SOA mixtures. Three different
parameterizations (Isaacman-VanWertz and Aumont, 2021; Mohr et al., 2019; Peräkylä et al., 2020) were used to convert the
SOA mass spectra to volatility distributions (VBS). The typical bulk volatilities of SOA from individual systems varied over
several orders of magnitude with the three parameterizations, as also reported by R. Wu et al. (2021) for the isoprene + NO$_3$
system. For all systems, the parameterization by Mohr et al. (2019) produced volatilities mainly in the ELVOC range, while
Peräkylä et al. (2020) produced narrow volatility distributions mainly in the SVOC range. Isaacman-VanWertz and Aumont
(2021) produced volatilities between the two other methods.

When used as input to a kinetic model, the VBS predicted by Mohr et al. (2019) reproduced the evaporation of α-
pinene ozonolysis generated SOA most accurately out of the three parameterizations. However, when simulating the
evaporation of the nitrate-initiated SOA, the Mohr et al. (2019) parameterization was found to generally significantly under-
predict the volatility and evaporation for all three precursors. The Isaacman-VanWertz and Aumont (2021) and Peräkylä et al.
(2020) parameterizations, on the other hand, generally over-predicted the volatility and evaporation for all the systems. The
results suggest that the Mohr et al. (2019) parameterization tends to substantially under-predict the volatilities of the organic
nitrates. This warrants a thorough re-evaluation of the parameter describing the magnitude of the vapor pressure reduction due
to nitrogen-containing functional groups, shifting the volatility distribution from the ELVOC towards the LVOC range. Taken
together, our experimental data and theoretical analyses suggest the SOA from the nitrate oxidation of α-pinene and isoprene
to contain a wide range of volatilities from the ELVOC to the SVOC range, with LVOCs dominating the composition. The
SOA from the nitrate oxidation of β-caryophyllene, on the other hand, seems to be dominated by volatilities in the ELVOC
and LVOC range. Our nitrate system represents SOA formation conditions where reactions between peroxy radicals and
between peroxy radical and NO$_3$ are favored and oligomers dominate. Further investigations to probe different ambient
conditions are warranted to complete the picture.



**Data availability**

The datasets are available upon request to the corresponding authors.

**Author contributions**

ELG, CW, DMB, IR and CM designed the study. Chamber experiments were carried out by ELG, CW, DMB, SH, AB and CM. Data analysis and interpretation was performed by ELG, CW, DMB, IR, and CM. ELG and CW wrote the manuscript, with input from all co-authors. All co-authors read and commented on the manuscript.

**Competing interests**

The authors declare that they have no conflict of interest.

**Acknowledgement**

Financial support from the European Union's Horizon 2020 research and innovation programme (project FORCeS under grant agreement No 821205), European Research Council (Consolidator grant INTEGRATE No 865799), and Knut and Alice Wallenberg foundation (Wallenberg Academy Fellowship projects AtmoRemove No 2015.0162, AtmoCLOUD No 2021.0169, and CLOUDFORM No 2017.0165) is gratefully acknowledged. Financial support from Max Planck society is also

gratefully acknowledged. We would also like to thank European Union's Horizon 2020 research and innovation program through the EUROCHAMP-2020 Infrastructure Activity under grant agreement no. 730997.



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
