# Peer review of "Volatility of aerosol particles from NO3 oxidation of various biogenic organic precursors"

_EGUsphere, 2022_

## Referee Comment (RC1)

In their work, "Volatility of aerosol particles from NO3 oxidation of various biogenic organic precursors," the authors present a detailed examination of the volatility distributions of SOA formed through BVOC+NO$_3$ reactions. Several different experimental methods are coupled with multiple formula-based parameterizations to generate a full picture of the distribution. In general, it is a very thorough and nice work, with cohesive conclusions and well-written presentation. NO$_3$ chemistry is an important and understudied topic, and there are some very interesting results here.

The formula-based vapor pressure parameterization approach used in this work is a topic with which I have some very specific expertise and interest, and I believe there are some specific major aspects that need to be re-examined, as discussed below. I do not think they will be hard to incorporate, but they will require some major revisions. I would be very happy to work with the authors directly and to answer any questions or clarifications, please do not hesitate to contact me at ivw@vt.edu.

> --Gabriel Isaacman-VanWertz.

(1) I don't think that the parameterization implemented in this work should be referred to as IvWA (also, I usually use an uppercase "V"). What is being implemented here is the Li et al. method, with modifications implemented as suggested by Isaacman-VanWertz and Aumont. The distinction being that we did not develop the coefficients or approach being used. Based on the best available structure-based estimation techniques, our proposed parameterization was actually to use the average of the modified Li method with the Daumit method, which is not included in the present work but relies heavily on SIMPOL like the Donahue and subsequent Mohr methods do. Modified Li et al. alone was shown to be higher volatility than the best structure-based estimation techniques, while Daumit (SIMPOL) tends to be lower. The average method is what is recommended by the PEACh scripts published as part of Isaacman-VanWertz and Aumont. I have actually updated PEACh to include the Mohr formulation am happy to send it along; I will update the git repository as soon as I get a chance, sorry it was not included in the first version.

(2) I would recommend trying the combined Li-Daumit method (which I would not object to being cited as IVWA, as it is a somewhat different approach than either alone). As noted above, it does the best job of mapping structure-based vapor pressure estimates onto formulas. This of course does not provide any information about how accurate the actual estimates are, which is a nice aspect of the present work, but it would still probably be the best parameterization to use for formulas. Below I have included comparison figures of the Mohr method with the modified Li method (essentially, the IvWA method in this work, though see comment 3 below), the Daumit method, and the Li-Daumit average recommended by Isaacman-VanWertz and Aumont. For comparison, the ~1200 formulas generated in Isaacman-VanWertz and Aumont are used, values are vapor pressure in units of atm (to convert to C* easily, just add roughly 9.5 orders of magnitude).

[Figure]

[Figure]

Figure RC1. Comparison of vapor pressures estimated for 1200 formulas using four parameterizations: Mohr et al. ("MHR" in manuscript), Modified Li et al. ("IvWA" in manuscript), Daumit et al., and average Li-Daumit (recommended by Isaacman-VanWertz and Aumont, using other Li modification, see comment 3). Values are in units of atm.

Generally, the Daumit method, which is based in part on SIMPOL coefficients like the Mohr method, is broadly near agreement with the Mohr method, while the modified Li method tends to predict higher vapor pressures because it is tuned to values from the EPI suite vapor pressure estimator, which tends to be higher than other methods. These features of the methods are consistent with the observations in the present work. IvWA is found to be generally too volatile, consistent with the known high-volatility tendency of the Li method, and the Mohr method is often found to be a bit too low volatility, consist with the Daumit method that is similarly based on SIMPOL (and the tendency of SIMPOL to predict lower volatilities than other methods).

The average Li-Daumit method is somewhat in between of course. Given that this method best captures structure-based estimates and might correct for some of the other methods in the present work, I would strongly suggest considering including it in the comparisons. As currently written, a lot of the results can be explained by the estimation methods, not inherently by the fact that the formula-based approaches implicitly ignore structure or by errors in the best available estimation methods. If Li-Daumit still fails (and it will, I'm sure, we're just not that good at these estimations

yet), we could better probe our capabilities of estimating these properties (Do we need structure? Or is even our structure-based estimation not good enough? See comment 5).

(3) I have a few minor concerns about how the modifications to Li et al. suggested by Isaacman-VanWertz and Aumont are actually presented, particularly in the discussion of the treatment of $NO_3$.

I find the formulation and discussion of Equation 4 a little confusing. What we originally proposed was $b_N= -2b_O$, with a different formulation of the actual equation (i.e., no subtraction term in the oxygen count). Using Equation 4 as written with $b_N=b_O$ is mathematically equivalent, but it is somewhat confusing because a reader may not realize the difference in the equation and may be confused by the two different equivalencies of $b_N$, especially given that you cite our work directly after stating $b_N=b_O$. I think it would be less confusing to keep the original Li formula and use $b_N=-2b_O$.

Similarly, I don't really agree with citing our work after the statement on line 194 that NO3 "thus reduces a compound's vapor pressure by 0.7732 orders of magnitude" – our work was agnostic to the magnitude of the decrease in vapor pressure, we simply noted that an $NO_3$ group is equivalent to an OH group in SIMPOL so should be able to be treated as such, as is down in the Mohr method. The magnitude of the effect, i.e., the coefficients, is defined by the model treatment of O in Li et al. (which is generally higher volatility as noted above). A similar issue arises on line 399, though if you referenced the method as a modified Li method I would not object to the characterization.

Note also, we actually proposed two different possible treatments of $NO_3$ – either directly setting $b_N=-2b_O$ as implemented in this work, or programmatically changing each $NO_3$ unit within the formula to an OH unit. In Isaacman-VanWertz and Aumont, we could not clearly establish either as better than the other and they both removed the apparent bias and are based on the same idea ($b_{NO3}=b_{OH}$). However, there is interestingly some difference, with the coefficient approach being used here yielding slightly higher vapor pressures than the conversion approach, see figure below of the 1200 formulas used in Isaacman-VanWertz and Aumont. I have included both approaches in the updated PEACh scripts that will be uploaded shortly.

[Figure]

Figure RC2. Comparison of vapor pressures estimated for 1200 formulas using two modifications to Li et al., one based on modified coefficients ("IvWA" in manuscript) and one changing NO3 units to OH (used by PEACh in average Li-Daumit method). Values are in units of atm.

(4)  In Figure 3, the three parameterizations are compared. It would be nice to see how these compare the VBS derived from the FIGAERO-CIMS distribution. In later figures, the parameterizations are compared to the TD and isothermal results, but there is no real comparison to the FIGAERO results, which could maybe be used as a sort of "truth". It would actually be interesting to try to tune a Li-like method that uses all the FIGAERO-CIMS thermograms to actually tune the coefficients. This thought is implicit in the efforts to tune $b_N$ by precursor (around line 420) - no single value works, but maybe there is a value that considers the CO interaction and maybe even some additional term.

(5)  Line 402 is an interesting point, that the strengths of these methods depend on their tuning. PRK is tuned for this, so of course it works. The others are tuned more generally, so they are more widely applicable, but maybe less accurate. See comment 4 above that maybe we can find a more universal tuning.

With that in mind, it would have been nice to see a little more in-depth dive into why these parameterizations do or don't capture different systems. Implicit in parameterizations is that each atom is associated with specific functional groups, so is this telling us something about differences in functionalities between systems? Do you see some formulas that desorb from FIGAERO at very different temperatures in each system? If so, that implies it is differences in structure that cannot be captured by formula alone. Or is it rather that formulas of some systems are being well represented and captured but other systems the formulas are different and are not being captured. For example, why do the parameterizations invert the volatility of the O3 and NO3 systems, is it just the volatility of $NO_3$ groups or are there other functionality differences contributing (e.g., do the distributions of their non-$NO_3$ components show the same inversion, or are they captured reasonably?)? These different possible outcomes would help answer the question of whether the issue in our vapor pressure estimation is that structural information is necessary (and inherently unavailable in MS data) or that our actual structure-based estimation methods are still not correct.

All of this is related to comment 4 above, using the FIGAERO data to probe what these parameterizations are actually telling us. If the authors are not interested in this problem, let me know, I would be happy to play with it if they would share the data. In general, I think the present data provide a great opportunity to understand and improve these parameterizations, but I'm not sure it is being done in this work or if the authors intend to examine this in the future.

All these comments above will have effects on discussion and figures throughout Sections 3.3 and 3.4, so I have not provided too much additional discussion or comments on these sections here.

Technical comments.

Overall, the paper is very nicely written, I have very few detailed technical comments.

Line 57: "number of studies…are much fewer" is not grammatically odd. Should be "number of studies…is lower" or "there are many fewer studies"

Line 70: I'm not sure the connection between the low-volatility of ON-containing HOMs and the 2.5 order drop from $NO_3$ follow from one another. I understand the connection the authors are pointing towards, but that same drop is true of OH groups, and there are plenty of SVOC and high volatility

alcohols. Assuming the 2.5 orders is correct, it takes a combination of functional groups to get to (E)LVOC.

Line 106-108: Run-on sentence

Line 206: Needs a comma before "referred"

Line 260-262: A bit of a complicated/run-on sentence

Figure 2: a-Pinene data is pretty hard to see, maybe use a darker green

Line 374: How is thermal decomposition subtracted out? And is it not then added back in as the presumed undecomposed mass? If not, that would remove a lot of low-volatility mass from the model, I would think. That is discussed somewhat, but I'm not completely clear how it is quantitatively being handles.

Line 387: "approach" should be plural or some other correction needs to be made

SI: It would be very helpful for the authors to share their data directly. In particular, what are the formulas observed in each oxidation experiment or system? For example, for Figure RC1 above I wanted to make the figure with the actual formulas used in this work, but they were not available, so I had to use those from Isaacman-VanWertz and Aumont.

---

## Author Comment (AC1)

*Response Letter to Referee #1*

The authors thank the reviewer for the careful review of our manuscript and helpful comments and suggestions, which significantly improved our manuscript. All the comments (in bold text) are addressed below point by point, with our response following in non-bold text and the corresponding revisions to the manuscript in blue. All updates of the original submission are tracked in the revised version.

**In their work, "Volatility of aerosol particles from NO3 oxidation of various biogenic organic precursors," the authors present a detailed examination of the volatility distributions of SOA formed through BVOC+NO3 reactions. Several different experimental methods are coupled with multiple formula-based parameterizations to generate a full picture of the distribution. In general, it is a very thorough and nice work, with cohesive conclusions and well-written presentation. NO3 chemistry is an important and understudied topic, and there are some very interesting results here.**

**The formula-based vapor pressure parameterization approach used in this work is a topic with which I have some very specific expertise and interest, and I believe there are some specific major aspects that need to be re-examined, as discussed below. I do not think they will be hard to incorporate, but they will require some major revisions.**

We thank the reviewer for the positive assessment of our manuscript. We are also thankful for offering to help with the implementation.

**(1) I don't think that the parameterization implemented in this work should be referred to as IvWA (also, I usually use an uppercase "V"). What is being implemented here is the Li et al. method, with modifications implemented as suggested by Isaacman-VanWertz and Aumont. The distinction being that we did not develop the coefficients or approach being used. Based on the best available structure-based estimation techniques, our proposed parameterization was actually to use the average of the modified Li method with the Daumit method, which is not included in the present work but relies heavily on SIMPOL like the Donahue and subsequent Mohr methods do. Modified Li et al. alone was shown to be higher volatility than the best structure-based estimation techniques, while Daumit (SIMPOL) tends to be lower. The average method is what is recommended by the PEACh scripts published as part of Isaacman-VanWertz and Aumont. I have actually updated PEACh to include the Mohr formulation am happy to send it along; I will update the git repository as soon as I get a chance, sorry it was not included in the first version.**

We thank the reviewer for clarifying the difference between the parameterization by Li, et al. (2016) and by Isaacman-VanWertz and Aumont (2021). We change now the naming of "IvWA" to "Li". We also thank the reviewer for suggesting to include the structure-based parameterization by Daumit, et al. (2013), as well as the average of the modified Li method and the Daumit method. We tested both and decided to add an updated modified version of the Daumit method which includes a term for nitrate functional groups in our manuscript, which is a very good addition. We added it in Methods (see text marked blue below), updated the Figs 3 and 4 as well as the discussion in the revised manuscript.

[revised manuscript text omitted]

**(2) I would recommend trying the combined Li-Daumit method (which I would not object to being cited as IVWA, as it is a somewhat different approach than either alone). As noted above, it does the best job of mapping structure-based vapor pressure estimates onto formulas. This of course does not provide any information about how accurate the actual estimates are, which is a nice aspect of the present work, but it would still probably be the best parameterization to use for formulas. Below I have included comparison figures of the Mohr method with the modified Li method (essentially, the IvWA method in this work, though see comment 3 below), the Daumit method, and the Li-Daumit average recommended by Isaacman-VanWertz and Aumont. For comparison, the ~1200 formulas generated in Isaacman-VanWertz and Aumont are used, values are vapor pressure in units of atm (to convert to C\* easily, just add roughly 9.5 orders of magnitude).**

[Figure]

[Figure]

**Figure RC1. Comparison of vapor pressures estimated for 1200 formulas using four parameterizations: Mohr et al. ("MHR" in manuscript), Modified Li et al. ("IvWA" in manuscript), Daumit et al., and average Li-Daumit (recommended by Isaacman-VanWertz and Aumont, using other Li modification, see comment 3). Values are in units of atm.**

Generally, the Daumit method, which is based in part on SIMPOL coefficients like the Mohr method, is broadly near agreement with the Mohr method, while the modified Li method tends to predict higher vapor pressures because it is tuned to values from the EPI suite vapor pressure estimator, which tends to be higher than other methods. These features of the methods are consistent with the observations in the present work. IvWA is found to be generally too volatile, consistent with the known high-volatility tendency of the Li method, and the Mohr method is often found to be a bit too low volatility, consist with the Daumit method that is similarly based on SIMPOL (and the tendency of SIMPOL to predict lower volatilities than other methods).

The average Li-Daumit method is somewhat in between of course. Given that this method best captures structure-based estimates and might correct for some of the other methods in the present work, I would strongly suggest considering including it in the comparisons. As currently written, a lot of the results can be explained by the estimation methods, not inherently by the fact that the formula-based approaches implicitly ignore structure or by errors in the best available estimation methods. If Li-Daumit still fails (and it will, I'm sure, we're just not that good at these estimations yet), we could better probe our

**capabilities of estimating these properties (Do we need structure? Or is even our structure-based estimation not good enough? See comment 5).**

We appreciate the reviewer sharing the thoughts and the work of comparing the Mohr method with the modified Li, the Daumit and the average of the modified Li and the Daumit methods. It is very interesting to see from Figure RC1 that the difference between the Daumit and the Mohr method become bigger for the low-volatility compounds, and that the Daumit method estimates lower volatilities of compounds with log$C^*$ between -20 and -10 using the Mohr method. It is consistent with our finding shown in the new Fig. 4a and 4e, and it explains very well why the Daumit method works less well than the Mohr method for the α-pinene + O$_3$ system. However, it is different for the nitrate systems. The Daumit method estimates much less ELVOC (new Fig. 3), and does a better job to estimate the VFR than the Mohr method (new Fig. 4).

From the new Fig. 4, we see that DMT tends to overestimate, and Li tends to underestimate the VFR in comparison with the VTDMA, thus the average Li-Daumit method will give the best estimation for the α-pinene + NO$_3$ and isoprene + NO$_3$ systems, but not for the β-caryophyllene + NO$_3$ system (see new Fig. S7 below). However, as both DMT and Li overestimate the VFR with the isothermal evaporation chamber for the α-pinene + NO$_3$ system and both underestimate the VFR for the β-caryophyllene + NO$_3$ system, the average Li-Daumit method would not improve the performance for these cases. To show the different performances of individual parameterizations, and meanwhile to keep Fig. 4 still readable, we decided to only add the modified Daumit method to the main manuscript.

[Figure]

**Figure S7. Comparison of the evaporation in the VTDMA system for the DMT (blue) and Li (green) methods, as well as their average (red).**

**(3) I have a few minor concerns about how the modifications to Li et al. suggested by Isaacman-VanWertz and Aumont are actually presented, particularly in the discussion of the treatment of NO3. I find the formulation and discussion of Equation 4 a little confusing. What we originally proposed was bN= -2bO, with a different formulation of the actual equation (i.e., no subtraction term in the oxygen count). Using Equation 4 as written with bN = bO is mathematically equivalent, but it is somewhat confusing because a reader may not realize the difference in the equation and may be confused by the two different equivalencies of bN, especially given that you cite our work directly after stating bN = bO. I think it would be less confusing to keep the original Li formula and use bN = -2bO.**
**Similarly, I don't really agree with citing our work after the statement on line 194 that NO3 "thus reduces a compound's vapor pressure by 0.7732 orders of magnitude" – our**

work was agnostic to the magnitude of the decrease in vapor pressure, we simply noted that an NO3 group is equivalent to an OH group in SIMPOL so should be able to be treated as such, as is down in the Mohr method.

The magnitude of the effect, i.e., the coefficients, is defined by the model treatment of O in Li et al. (which is generally higher volatility as noted above). A similar issue arises on line 399, though if you referenced the method as a modified Li method I would not object to the characterization. Note also, we actually proposed two different possible treatments of NO3 – either directly setting bN = -2bO as implemented in this work, or programmatically changing each NO3 unit within the formula to an OH unit. In Isaacman-VanWertz and Aumont, we could not clearly establish either as better than the other and they both removed the apparent bias and are based on the same idea (bNO3 = bOH). However, there is interestingly some difference, with the coefficient approach being used here yielding slightly higher vapor pressures than the conversion approach, see figure below of the 1200 formulas used in Isaacman-VanWertz and Aumont. I have included both approaches in the updated PEACh scripts that will be uploaded shortly.

[Figure]

**Figure RC2. Comparison of vapor pressures estimated for 1200 formulas using two modifications to Li et al., one based on modified coefficients ("IvWA" in manuscript) and one changing NO3 units to OH (used by PEACh in average Li-Daumit method). Values are in units of atm. Modified Li using bN = -2bO**

The focus of this paper is organic nitrates, and we would like to clearly see the impacts of nitrate functional groups also in the equations. We think using $b_N = -2b_O$ is not as clear as the mathematically equivalent Eq. 9 in the revised manuscript, and would therefore prefer to keep the equation as is:

$$\log_{10} C^* = (n_0 - n_C)b_C - (n_O - 3n_N)b_O - 2\frac{n_C n_O}{n_C + n_O}b_{CO} - n_N b_N$$

With new Eq. 9, the NO3 group is clearly separated from the other functional groups containing oxygen atoms, and $b_N$ ($b_{\text{-NO3}}$) = $b_O$ ($b_{\text{-OH}}$). In particular, when we tune this parameterization for NO3 systems, we just want to tune the $b_N$ value, rather than the treatment to other functional

groups containing oxygen atoms ($b_O$). Due to the same reason, we didn't present the second method (converting -NO$_3$ to -OH) for the modified Li and the modified Daumit methods. However, we do agree to delete "thus reduces a compound's vapor pressure by 0.7732 orders of magnitude ($b_N = b_O$ (Isaacman-VanWertz and Aumont, 2020)).", as suggested.

We also thank the reviewer for bringing up the differences induced by the two modifications (Figure RC2). On the one hand, it is not surprising because without converting -NO$_3$ to -OH the b values are obtained by optimization using a database containing CHON compounds, while by converting -NO$_3$ to -OH, the b values are from a database containing CHO compounds. On the other hand, it also hints at differences in the "actual" b values for -OH and -NO$_3$.

**(4) In Figure 3, the three parameterizations are compared. It would be nice to see how these compare the VBS derived from the FIGAERO-CIMS distribution. In later figures, the parameterizations are compared to the TD and isothermal results, but there is no real comparison to the FIGAERO results, which could maybe be used as a sort of "truth". It would actually be interesting to try to tune a Li-like method that uses all the FIGAERO-CIMS thermograms to actually tune the coefficients. This thought is implicit in the efforts to tune bN by precursor (around line 420) - no single value works, but maybe there is a value that considers the CO interaction and maybe even some additional term.**

Thank you for this comment. Indeed, the thermograms of individual ions from the desorption in the FIGAERO-CIMS provide another means to assess volatilities. As we have shown earlier, e.g. in a previous publication on this dataset (Wu, et al. 2021), especially for complex mixtures the quantitative assessment of $C^*$ is challenging using FIGAERO-CIMS thermograms, to say the least. We therefore slightly disagree with the reviewer's statement of using the FIGAERO thermograms as a sort of "truth". Schobesberger, et al. (2018) have developed a model framework to reproduce the shape of FIGAERO-CIMS thermograms largely based on the Hertz–Knudsen equation (Cappa, et al. 2007; Hertz 1882). The geometry of the FIGAERO inlet is far more complex than the TD, and the evaporation of compounds from the gradually heated filters includes more complex kinetic processes than e.g., the corresponding evaporation within a TD setup, which would at present add a large uncertainty in the interpretation of the thermograms.

Whereas this model could potentially be used to find $C^*$ of a certain compound via tuning the modelled to the measured thermogram shape, Schobesberger, et al. (2018) also discuss uncertainties related to the fact that several solutions may be possible to reproduce a certain thermogram. This is especially relevant for multi-component systems and for compounds where no initial guesses for $C^*$ are available from the literature. Using the results from the different parameterizations might provide such initial guesses, of course. We feel like the extensive analysis that would be required to fully explore the parameterization – thermogram relationship is beyond the scope of this paper, but presents an interesting option for a new study.

In Fig. AC2, we plot the maximal desorption temperature ($T_{max}$) of the individual compounds observed with the FIGAERO and compare the $T_{max}$ to the predicted evaporation temperature in TD at which 50 % and 99 % of compounds should evaporate according to the kinetic model. It shows clearly the differences in the evaporation with the FIGAERO and TD. Using calibration standards for extracting vapor pressure of measured compounds (Bannan, et al. 2019) might be another option, but we haven't done such calibration during the experiments.

[Figure]

Fig. AC2. The TD temperatures at which 50 % (dashed line) or 99 % (dotted line) of species with a given $C^*$ are predicted to evaporate for (a) MHR, (b) Li and (c) PRK according to the kinetic model. Grey shading indicates regions where the majority of the species with a given $C^*$ are expected to be in the particle phase, and pink (+ blue) shading regions with gas phase dominating the partitioning within our VTDMA setup. The markers represent the mean maximum desorption temperature ($T_{max}$) within the FIGAERO-CIMS as a function of the $C^*$ values for α-pinene ozonolysis (purple) and the nitrate oxidation of α-pinene (green), isoprene (blue) and β-caryophyllene (orange). The marker sizes are proportional to the mass fractions observed in the studied systems. The blue areas mark the range of $C^*$ which is sensitive towards iso-thermal evaporation at room temperature with our setup, based on the results shown in Figs. 4 and 5 in the original manuscript.

**(5) Line 402 is an interesting point, that the strengths of these methods depend on their tuning. PRK is tuned for this, so of course it works. The others are tuned more generally, so they are more widely applicable, but maybe less accurate. See comment 4 above. With that in mind that maybe we can find a more universal tuning, it would have been nice to see a little more in-depth dive into why these parameterizations do or don't capture different systems. Implicit in parameterizations is that each atom is associated with specific functional groups, so is this telling us something about differences in functionalities between systems? Do you see some formulas that desorb from FIGAERO at very different temperatures in each system? If so, that implies it is differences in structure that cannot be captured by formula alone. Or is it rather that formulas of some systems are being well represented and captured but other systems the formulas are different and are not being captured. For example, why do the parameterizations invert the volatility of the O3 and NO3 systems, is it just the volatility of NO3 groups or are there other functionality differences contributing (e.g., do the distributions of their non-NO3 components show the same inversion, or are they captured reasonably?)? These different possible outcomes would help answer the question of whether the issue in our vapor pressure estimation is that structural information is necessary (and inherently unavailable in MS data) or that our actual structure-based estimation methods are still not correct.**

**All of this is related to comment 4 above, using the FIGAERO data to probe what these parameterizations are actually telling us. If the authors are not interested in this problem, let me know, I would be happy to play with it if they would share the data. In general, I think the present data provide a great opportunity to understand and improve these parameterizations, but I'm not sure it is being done in this work or if the authors intend to examine this in the future.**

**All these comments above will have effects on discussion and figures throughout Sections 3.3 and 3.4, so I have not provided too much additional discussion or comments on these sections here.**

The reviewer has brought up a very good question: whether structural information is necessary for estimating vapor pressure. We think the answer is, very clearly, yes. This is why e.g., in Mohr, et al. (2019), the $b_O$ value was tuned for -OOH functional groups rather than the -OH and =O functional groups for HOMs, and why e.g. in Isaacman-VanWertz and Aumont (2021), the $b_N$ values are adjusted to be the same as of -OH to better represent -$NO_3$ functional groups, rather than amines and amides etc. This could also be the reason why in our study MHR works better for the ozonolysis system, but DMT works better for the nitrate systems. Considering that the nitrate group contribution term in two methods is similar and the $b_N$ value is also similar (-2.5 in the Mohr method, and -2.23 in the Daumit method), the way how these two parameterizations treat other functional groups must play a role. The Daumit method assumes all oxygen atoms except those from -$NO_3$ are from -OH, =O functional groups, while the Mohr method was tuned for -OOH functional groups and the $b_O$ (=0.2) is much smaller than $b_{-OH}$ (= -2.23) and $b_{=O}$ (= -0.935) in DMT. It is possible that α-pinene + $O_3$ system has more -OOH and α-pinene + $NO_3$ system has more -OH and =O. However, we cannot exclude potential impacts from e.g. the carbon-carbon interaction term ($b_C$) and the carbon-oxygen nonideality ($b_{CO}$), as these two systems also have different carbon and oxygen numbers.

We added some discussion in the revised manuscript:

"…the parameterizations. The MHR and Daumit parameterizations are similar in the $b_N$ values but different in the way they treat other functional groups containing oxygen atoms. The Daumit method assumes all oxygen atoms except those from -$NO_3$ are form -OH and =O functional groups, while the Mohr method was tuned for -OOH functional groups and the $b_O$ (=0.2) is much smaller than $b_{-OH}$ (= -2.23) and $b_{=O}$ (= -0.935) in DMT. As MHR works better for the α-pinene + $O_3$ system and DMT works better for the α-pinene + $NO_3$ system, it is possible that the α-pinene + $O_3$ system has more -OOH and the α-pinene + $NO_3$ system has more -OH and =O."

In Fig AC3, $T_{max}$ of the compounds with the same molecular formula and from two different systems (isoprene + $NO_3$ and α-pinene + $NO_3$) are shown. Clearly, if they have the same/a similar structure (e.g. $C_{10}H_{15}NO_{3+n}$), their $T_{max}$ should be similar given similar experimental conditions, while very different structures (e.g. $C_{10}H_{16}N_2O_{6+n}$) could lead to very different $T_{max}$. However, such common compounds are not ubiquitous, as isoprene, α-pinene, β-caryophyllene have different carbon numbers, and numbers of double bonds. Thus, an overall check of functional groups for different systems based on FIGAERO thermograms is not possible. As we didn't have enough information about structure and then to distinguish the different functional groups, the question "whether our actual structure-based estimation methods are still not correct" is beyond the scope of this paper.

We are, however, happy to share the experimental data and will make it publicly available through the Bolin Centre database for the final version of the manuscript. We look forward to any further insights stemming from this dataset.

[Figure]

Fig. AC3 Maximum desorption temperature ($T_{max}$) of compounds from isoprene + NO$_3$ and α-pinene + NO$_3$ experiments. numO is the number of oxygen atoms from other functional groups than nitrate groups.

**Technical comments.**

**Overall, the paper is very nicely written, I have very few detailed technical comments.**
**Line 57: "number of studies…are much fewer" is not grammatically odd. Should be "number of studies…is lower" or "there are many fewer studies"**
Thank you. The original sentence "The number of studies on NO$_3$ oxidation of biogenic VOCs (BVOCs) are much fewer" is revised to "There are much fewer studies on NO$_3$ oxidation of biogenic VOCs (BVOCs)".

**Line 70: I'm not sure the connection between the low-volatility of ON-containing HOMs and the 2.5 order drop from NO3 follow from one another. I understand the connection the authors are pointing towards, but that same drop is true of OH groups, and there are plenty of SVOC and high volatility alcohols. Assuming the 2.5 orders is correct, it takes a combination of functional groups to get to (E)LVOC.**
The reviewer is absolutely right that for HOMs containing nitrate groups, not only nitrate groups but also other functional groups lower the volatility. However, one nitrate group contains three oxygen atoms, which, if they came from other functional groups, could be equal to 3 -OH or =O, or 1.5 -OOH groups. Thus, we think if these HOMs with nitrate groups fall within the category of ELVOC, the drop in their volatility due to the nitrate group must be rather big, and 2.5 is a reasonable value.

**Line 106-108: Run-on sentence**
The original sentence is revised to "In the O$_3$ experiments, O$_3$ was generated by photolysis of air. Similar to the NO$_3$ experiments, the chamber was first humified and then O$_3$ and α-pinene with the desired levels were added to the chamber volumetrically."

**Line 206: Needs a comma before "referred"**
Accepted.

**Line 260-262: A bit of a complicated/run-on sentence**

The original sentence is revised to "The VTDMA plots (Fig. 1a) show that particles formed through nitrate oxidation consistently display a lower VRF at given temperatures than the particles from ozonolysis, hence have higher bulk volatility."

**Figure 2: a-Pinene data is pretty hard to see, maybe use a darker green**

We changed the α-pinene colour to grey in the background.

[Figure]

**Figure 2** Measured volatility of SOA from isoprene (a-c, blue curves) and β-caryophyllene (d-f, red curves) oxidation by NO₃. The corresponding data from α-pinene oxidation are marked in grey. The left column (a, d) presents the mean VFR evolution during the VTDMA temperature ramps, the middle column (b, e) the isothermal evaporation, and the right column (c, f) mean thermograms measured by the FIGAERO-CIMS. The VFR at room temperature (a, d) is estimated based on predicted evaporation at room temperature (b, e). For the VTDMA experiments, the VFR values correspond to the mean of three and two individual experiments for isoprene and β-caryophyllene, respectively (see Table 1), where the uncertainty bars correspond to the sum of the standard deviation of the VFR and the error of the two DMPS systems within the VTDMA (see Sect. 2.2.3).

**Line 374: How is thermal decomposition subtracted out? And is it not then added back in as the presumed undecomposed mass? If not, that would remove a lot of low-volatility mass from the model, I would think. That is discussed somewhat, but I'm not completely clear how it is quantitatively being handles.**

The thermal decomposition subtraction is described in Wu et al. (2021). It is then not added back in because the volatility calculated based on the formula of their thermal fragmentation compounds is much higher than the thermal decomposition compounds of their parent compounds, thus it will lead to discrepancy in further calculation of the bulk volatility and the simulation of evaporation. We listed the signal fraction of the thermal decomposition compounds for each system and added some sentences in the revised manuscript to clarify this and the potential effects on results.

"It is worth noting that with the FIGAERO-CIMS, subtraction of the thermal decomposition compounds may also induce uncertainties. Thermal fragmentation contributed 5%–27%, 1%–4%, and 10%–23% of the total organic signal of the isoprene, α-pinene, and β-caryophyllene nitrate SOA, respectively, and 3–13% for α-pinene ozonolysis SOA. Since thermal decomposition due to dehydration or decarboxylation reactions normally occurs at temperatures higher than 120 °C (Buchholz, et al. 2020; Stark, et al. 2017) and the parent compounds of the decomposing molecules have an even higher thermal desorption temperature, we can conclude that the compounds we remove are of very low volatility. Unfortunately, their volatility could not be identified with the parameterization based on the formula of the thermal decomposition compounds, as they are much smaller than the parental compounds and thus have much higher volatilities. We subtracted these compounds for the further calculation of the bulk volatility and the simulation of evaporation, but we also note that it may lead to decreases in the simulated VFR in the higher temperature range (> 120 °C) in the VTDMA, thus it could be a potential reason for the discrepancies between the calculated VFR and the measured VFR with the VTDMA. In addition, as α-pinene nitrate SOA exhibits much less thermal fragmentation compared to isoprene and β-caryophyllene, it could also explain partly the different performance of parameterization in simulating the evaporation in the VTDMA in different systems."

**Line 387: "approach" should be plural or some other correction needs to be made**
Change to "as well as experimental approaches".

**SI: It would be very helpful for the authors to share their data directly. In particular, what are the formulas observed in each oxidation experiment or system? For example, for Figure RC1 above I wanted to make the figure with the actual formulas used in this work, but they were not available, so I had to use those from Isaacman-VanWertz and Aumont.**
We are very happy to share the experimental data. As mentioned above we will make it publicly available through the Bolin Centre database for the final version of the manuscript.

---

## Author Comment (AC2)

*Response Letter to Referee #2*

The authors thank the reviewer for the careful review of our manuscript and helpful comments and suggestions. All the comments (in bold text) are addressed below point by point, with our response following in non-bold text and the corresponding revisions to the manuscript in blue. All updates of the original submission are tracked in the revised version.

**General comments:**

**In this work, "Volatility of aerosol particles from NO3 oxidation of various biogenic organic precursors," the authors examine analytically combining the use of different experiment setups, as a thermodenuder, an isothermal chamber and a FIGARO-CIMS system, formula-based parameterizations and an evaporation model in order to try to estimate volatility from the NO3 oxidation of a-pinene, isoprene and b-caryophyllene, a topic that is very understudied and scarce information exists about these volatility estimations. Volatility estimation is a topic that requires a lot of effort, the combination of different experimental and modeling techniques and sometimes the results about it can be failing or varied. This happens because of the complexity of the mechanisms of aging and reactions that happen in the atmosphere that makes the fate of the precursors highly uncertain. Also using modeling techniques there are many assumptions that have to be made in order to infer volatility that make the problem even more complex. The authors manage to use rigorously a combination of methods and finally provide us with a descriptive analysis of the volatility distributions and composition of SOA formed from nitrate oxidation for the three precursors. In the end the formula-based parameterizations were re-evaluated and showed that improvements in their description have to be incorporated in the future. I find this work very valuable and almost ready to be published while a few comments could be addressed a bit better and some technical corrections could be performed in order to be clearer.**

We thank the reviewer for the positive assessment of our manuscript.

**Specific comments:**

**In Table 1 are presented the experiments performed for this work and while most of the results of them are presented experiment 4 is not used in the analysis. It is only referred to figure S2 when trying to show the sensitivity of different initial mass loadings from the nitrate oxidation of a-pinene. Is there any specific reason to not in the main figures as well? Also in the table there are many missing values, this could be explained in the bottom why is it happening as a footnote. Also while the nitrate experiments of a-pinene are 5, is there any reason there are 3 for isoprene and 2 for b-caryophyllene. Probably some description for the decision of the design of these experiments could be helpful in the main article. Finally for the isothermal evaporation experiments while most experiments finish at 240 min some stop after 150 min, this is probably because the experiment had to stop or there was equilibrium and no extra evaporation was noticed after that?**

For the evaporation simulation we only used experiments for which we have FIGAREO-CIMS data (so we included exp. 3 and 4 solely to illustrate the mass dependence in the isothermal evaporation chamber for α-pinene + NO₃). Unfortunately, there are two experiments where we have FIGAERO-CIMS data but miss either VTDMA (exp. 5) or isothermal evaporation chamber data (exp. 8). Same for the α-pinene ozonolysis experiments, see exps 1 and 2. In the revised paper, we added this information in the caption.

"…evaporation (experiments marked with hyphens are not simulated due to either missing data about the molecular composition or experimental evaporation data)."

**The evaporation model that is used for the analysis from Riipinen, et al. (2010) depends on many parameters and some work has been done to investigate the sensitivity of them as it is done in Figure S3 where there is used a different value for the vaporization enthalpy of 70kJ/mol, different accommodation coefficients of 0.01 and 0.1 compared to unity of the base case as well as a mass-dependent diffusion coefficient. This analysis has been focused only on the ozonolysis of a-pinene though that is not the centre of interest in this work and in any case it showed to have better comparisons with the measurements. The b-caryophyllene nitrate oxidation products seem to have a more bimodal volatility distribution that make it more difficult to capture its "fingerprint" with the VFR in the thermodenuder and isothermal evaporation chamber, it could be probably fruitful to try to see for the three different parameterizations used how changing dHvap (with values of 70 kj/mol but also higher even 150 kj/mol), am (0.01 and 0.1) and mass-dependent diffusion coefficients would change these thermograms. This could be repeated for the isoprene case for a more broad and complete picture for the sensitivity to these parameters.**

Thank you for this suggestion. We have now conducted similar sensitivity tests as done for α-pinene + O₃ in the original manuscript for the other investigated systems (see Figs. below, which have been added as supplementary figures in the revised manuscript). These figures demonstrate that the sensitivities are rather similar and independent of the exact chemical system.

[Figure]

**Figure S3** Sensitivity study for the thermodenuder- (upper row, panels a, b, c and d) and isothermal evaporation model (lower row, panels e, f, g and h) based on α-pinene ozonolysis with a fixed vaporization enthalpy, ΔH$^{VAP}$ = 70 kJ/mol (yellow), reduced accommodation (αm, set to be either 0.1 or 0.01 (pink)), and a mass-dependent diffusion coefficient (cyan). The original model output is presented in purple together with the measurements displayed in grey.

[Figure]

**Figure S4** Sensitivity study for the thermodenuder- (upper row, panels a, b, c and d) and isothermal evaporation model (lower row, panels e, f, g and h) based on α-pinene + NO$_3$ with a fixed vaporization enthalpy, ΔH$^{VAP}$ = 70 kJ/mol (yellow), reduced accommodation (αm, set to be either 0.1 or 0.01 (pink)), and a mass-dependent diffusion coefficient (cyan). The original model output is presented in purple together with the measurements displayed in grey.

[Figure]

Figure S5 Sensitivity study for the thermodenuder- (upper row, panels a, b, c and d) and isothermal evaporation model (lower row, panels e, f, g and h) based on Isoprene + $NO_3$ with a fixed vaporization enthalpy, $\Delta H^{VAP} = 70$ kJ/mol (yellow), reduced accommodation ($\alpha m$, set to be either 0.1 or 0.01 (pink)), and a mass-dependent diffusion coefficient (cyan). The original model output is presented in purple together with the measurements displayed in grey.

[Figure]

Figure S6 Sensitivity study for the thermodenuder- (upper row, panels a, b, c and d) and isothermal evaporation model (lower row, panels e, f, g and h) based on β-caryophyllene + $NO_3$ with a fixed vaporization enthalpy, $\Delta H^{VAP} = 70$ kJ/mol (yellow), reduced accommodation ($\alpha m$, set to be either 0.1 or 0.01 (pink)), and a mass-dependent diffusion coefficient (cyan). The original model output is presented in purple together with the measurements displayed in grey.

**In Figure 3 the authors show the volatility distributions in the form of VBS as derived from the three different parameterizations that are examined, while someone can see the composition in ELVOCs, LVOCs and SVOCs it would be useful here to also add in the figure somewhere what is the average C\* from each parameterization for each case of precursor. This would make it clearer understanding which parameterization shows higher or lower volatilities for each case.**

Thank you for this suggestion, we have added this information to the legend of the new Fig. 3 in the revised manuscript.

**Technical corrections:**

**Table 1: In footnote change S5 to S4 (there is no figure S5)**

Accepted.

**Line 227: Change 240 s to 240 min.**

Accepted.

**Line 228: model inputs instead of input**

Accepted.

**Line 229: Change concentrations estimated to concentration estimates to be clearer**

Accepted.

**Line 267: change to "agree with the VTDMA data"**

Accepted.

**Line 327: A bit complicated sentence, rewrite clearer**

Change to "Finally, with the PRK parametrization, the VBS distributions of all systems are similar and in a narrow range between -3 to 1, which suggests that this parameterization is the least sensitive."

**In Figure 4 I think the shading should change because right now it is not very easy and clear to see the differences, especially in the green shades.**

We have tried to address this problem by changing the shades of the different colours slightly. Additionally, we have separated the different precursors and oxidants to one figure each in the supplementary (new Fig. S8-11), presenting one VBS per subplot.

[Figure]

**Figure S8. Volatility distributions in the form of VBS as derived from the molecular composition for α-pinene ozonolysis, presented for each parameterization separately; (a) MHR: Mohr et al., 2019 (b) DMT: Daumit et al., 2013 (c) Li: modified Li et al., 2016 and (d) PRK: Peräkylä et al., 2020).**

[Figure]

**Figure S9. Volatility distributions in the form of VBS as derived from the molecular composition for the nitrate oxidation of α-pinene, presented for each parameterization separately; (a) MHR: Mohr et al., 2019 (b) DMT: Daumit et al., 2013 (c) Li: modified Li et al., 2016 and (d) PRK: Peräkylä et al., 2020).**

[Figure]

**Figure S10. Volatility distributions in the form of VBS as derived from the molecular composition for the nitrate oxidation of isoprene, presented for each parameterization separately; (a) MHR: Mohr et al., 2019 (b) DMT: Daumit et al., 2013 (c) Li: modified Li et al., 2016 and (d) PRK: Peräkylä et al., 2020).**

[Figure]

**Figure S11. Volatility distributions in the form of VBS as derived from the molecular composition for the nitrate oxidation of β-caryophyllene, presented for each parameterization separately; (a) MHR: Mohr et al., 2019 (b) DMT: Daumit et al., 2013 (c) Li: modified Li et al., 2016 and (d) PRK: Peräkylä et al., 2020).**

**Line 378 Change Though to Although**

Accepted.

**Line 387 something is missing, rewrite "and experimental approach developed by the authors"**

Change to "than that estimated by other parameterisations, e.g. SIMPOL (Pankow and Asher, 2008) and PRK, as well as that obtained by experimental approaches".

**Line 399: Change to "an hydroxyl"**

"a hydroxyl group" should be correct.

**Line 408: Change to "of the observations"**

Change to "all three systems to get predictions closer to the observations."

**Line 450: Taken together, change the expression, suggest that the SOA … contain, rewrite a bit the sentence.**

The original sentence is revised to "By comparing the volatility information from different parameterizations with that derived from temperature-dependent evaporation in VTDMA and isothermal evaporation chambers, our study suggests…"

**In Supplementary:**

**Figure S1: Change Narve, 2019 to NArVE campaign 2019 as it is mentioned in the article or This work**

Accepted.

**Figure S3: Change blue to cyan (you use cyan color)**

Accepted.

**Figure S3: Describe a,b,c,d,e,f in legend.**

The figure caption is revised to "Sensitivity study for the thermodenuder- (upper row, panels a, b, c and d) and isothermal evaporation model (lower row, panels e, f, g and h) based on α-pinene ozonolysis with a fixed vaporization enthalpy, $\Delta H^{VAP}$ = 70 kJ/mol (yellow), reduced accommodation (αm, set to be either 0.1 or 0.01 (pink)), and a mass-dependent diffusion coefficient (cyan). The original model output is presented in purple together with the measurements displayed in grey."

References

Riipinen, Ilona, et al.

> 2010 Equilibration time scales of organic aerosol inside thermodenuders: Evaporation kinetics versus thermodynamics. Atmospheric Environment 44(5):597-607.

---

## Author Response (AR2)

*Response Letter to Referee #1*

The authors thank the reviewer for the careful review of our response and the revised manuscript and helpful comments. All the comments (in bold text) are addressed below point by point, with our response following in non-bold text and the corresponding revisions to the manuscript in blue. All updates of the original submission are tracked in the revised version.

**In their manuscript "Volatility of aerosol particles from NO3 oxidation of various biogenic organic precursors", the authors present a revised discussion of volatility of observed products classified by molecular formula, comparing four volatility parameterizations and two volatility measurement approaches. This is a valuable contribution and is nearly ready for publication once an issue in the Daumit parameterization is addressed (see major comment 1).**

**I think the reviewers for their thoughtful responses and appreciate their detailed discussion of the C\*-T_D relationship from FIGAERO, which I know has been the subject of a lot of work since I last dealt with it. I look forward to looking at their data to see what else we can learn.**

We thank the reviewer for the positive assessment of our response and the revised manuscript.

**Major comments:**

**(1) I am not convinced the implementation of the Daumit method is really correct. The assumption that the only source of DBE is carbonyl (and nitrate) is not necessarily correct, since rings add DBE and may be preserved in a-pinene and B-caryophyllene products. This is typical for the Daumit method, but the impact of rings should probably be mentioned explicitly, since a-pinene and B-cary products can very possibly have rings preserved. More importantly though, this issue can be dangerous given the equations as written. If there are other sources of DBE (e.g., rings), the number of carbonyls Eq. 6 will produce can actually return more carbonyls than there are functional groups. For example, the a-pinene product C10H14N2O8 (observed in this work as shown in Fig. AC3) is calculated to have 3 carbonyls (10-14/2-2/2+1 = 3), though in reality there are likely two only two non-nitrate other functional groups. (and probably a preserved ring). For other cases, for instance C7H9NO5 which GECKO-A predicts as a product, equation 6 estimates 3 carbonyls (7-9/2-1/2+1 = 3), again impossible as there are only two non-nitrate oxygens. Another example is C9H12O3, a GECKO-A predicted non-nitrate product that equation 6 estimates contains 5 carbonyls despite having only 3 oxygens. This issue probably applies to a minor fraction of formulas, but is probably present to some degree.**

**Part of the issue comes from the definition of n_carbonyl, which is defined here as DBE-n_NO3. For one thing, I don't think the double bond in the nitrate actually matters in terms of carbonyl groups because it doesn't come at the expense of two hydrogens on the arbon backbone like the carbonyl does (in other words, it is not giving us any information about double-bonded carbons, which is used to estimate carbonyl number). However, the**

**nitrate group itself does come at the expense of a hydrogen, so it is more accurately considered as n_carbonyl = DBE = n_C-(n_H+n_N)/2+1. The end equation is actually the same, so maybe it doesn't matter, but I'm not sure it is actually correct as written. More importantly, though, is the excess DBE problem. I think you have to write it as n_carbonyl = min(n_C-(n_H+n_N)/2+1 , n_O-3*n_N). Basically, you can't have more carbonyls than you have functional groups. This issue is inherent in this formulation of the Daumit method, which was originally developed to estimate carbon number from AMS data. I think it has to be implemented in the form log c\* = log a + b_0 + b_C\*n_C + b_OH\*n_OH + b_=O\*n_=O + b_NO3\*n_NO3. I'm not sure where that leaves you with Eq. 8, but this approach would still allow the authors to tune the b_NO3 as done in Table 2.**

**This change probably won't have significant influence on your figures or conclusions, but should be implemented prior to publication.**

The reviewer brings up interesting points. Indeed, the definition of DBE includes the contribution of double bonds as well as rings, i.e. defines the degree of carbon saturation. Hence, as pointed out by the reviewer, by using the Daumit method, which determines the number of carbonyl groups based on DBE, and by us modifying it to accommodate nitrate groups, the number of carbonyls may be overestimated. Whereas the Daumit method provides some information on molecular structure, it is still a method based on molecular composition, and thus cannot assign functional groups, e.g. rings, 100% correctly.

The reviewer brings up a few examples of compounds (based on GECKO-A), where the number of oxygen atoms available for carbonyl functional groups is less than the number of carbonyl groups predicted by the modified Daumit method. We have checked all identified molecular formulae in our experiments and find that this applies to 48 (out of 415), 85 (out of 425) and 117 (out of 366) of compounds (making up 2-3%, 29-45% and 35-41% of signal) for isoprene, $\alpha$-pinene and $\beta$-caryophyllene, respectively. We however abstain from introducing a minimum condition in the parametrization, as introducing a condition for the maximum number of carbonyls never exceeding the total number of functional groups would improve the overestimation of carbonyl groups, but not entirely solve this issue. For instance, consider the compound $C_{20}H_{32}N_2O_8$ from the $\alpha$-pinene + $NO_3$ system:

Based on the modified Daumit method as presented in the paper we would for $C_{20}H_{32}N_2O_8$ get four carbonyl groups that contain non-nitrate oxygen atoms. With n_carbonyl = nO-3*nN (i.e. limiting the number of carbonyl groups based on available non-nitrate oxygen), we would obtain two carbonyl groups. This would then equally not represent the actual structure of the molecule well, as neither the four rings, nor the peroxide group are represented correctly. We therefore feel like adding additional conditions would not improve the parametrization. Instead, we do see the

possibility of running chemical kinetic models to get more detailed structure information, potentially improve the application of DMT, and get a better estimation of the volatility. Such a rather comprehensive endeavor is, however, outside the scope of this study.

We added the following discussion about the impacts of rings to the revised manuscript:

"As MHR works better for the α-pinene + $O_3$ system and DMT works better for the α-pinene + $NO_3$ system, it is possible that the α-pinene + $O_3$ system has more -OOH and the α-pinene + $NO_3$ system has more -OH and =O. However, it is worth noting that as the definition of DBE includes double bonds and rings, and we cannot exclude contributions of oxidation products with rings retained in the α-pinene and β-caryophyllene systems, we note that n_carbonyl represents an upper limit."

The reviewer also brings up another issue: the treatment of the nitrate group in the calculation of DBE. Whereas we agree with the reviewer that the number of double bonds in the nitrate group does not give information about double-bonded carbon and with that the number of carbonyls in a molecule, the DBE formulation still includes all double bonds in all functional groups. As also pointed out by the reviewer, the subtraction of one double-bond equivalent per nitrate group from DBE yields the same number of carbonyls as treating a terminal nitrogen as a hydrogen or –OH. These are just different ways of thinking about a molecule's structure. We therefore keep the formulation as is. In order to make this clear to readers, we have added the following sentences in the manuscript:

"We note that including the nitrate group in the calculation of $n_{=O}$ can be critically discussed, as the nitrate group does not yield any information on double-bonded carbons. However, treating the N of the nitrate group as a hydrogen in the DBE formulation ($C + 1 - (\frac{H}{2} + \frac{N}{2})$), yields the same final formula for $n_{=O}$ (Eq. 6)."

**(2) The results in figure AC3 are very interesting. As the authors note in their reply, the impact of structure is of course quite important (trying to quantify it was in fact the main goal of Isaacman-VanWertz and Aumont), so it is interesting to see here. In particular, the C10H16N2O6 desorption temperatures are very different between the systems. The structures are of course quite different, since the isoprene structure must be a dimer, but it is nevertheless fascinating to see such a large difference, since the functional groups are probably very similar, so it is probably an interaction term that will be hard to capture. I look forward to exploring the data once posted!**

Thank you!

Technical comments:

**Line 59 is still not correct. Should be "there are many fewer", not there are "much fewer"** Accepted.

**Line 154. Should be "Samples were" not "Samples was"**

Accepted.

**Line 363. I would clarify a little and say that "DMT leads to higher VFR than the observation \*at long evaporation time\*...", since until then it is basically on time of observations, more or less like MHR.**

We revised the sentence to make it more clear:

"The calculated evaporation of SOA from α-pinene ozonolysis in the VTDMA (Fig. 4a) using MHR and DMT reproduces the evaporation well, while the other two parameterizations, Li and PRK, result in lower VFR compared to the measurements. The overall performance is comparable for the isothermal evaporation data (Fig. 4e), i.e., MHR reproduces the evaporation best, while Li and PRK underestimate VFR, however, DMT overestimate the evaporation over time."

**Line 427-428. This point about the C-O non-ideality is a very good one I think I had missed before. It is probably the reason for the difference in the Li modification between using the coefficient change (as used here) and the algorithmic conversion of NO3 to OH (as used in Isaacman-VanWertz and Aumont). The NO3 oxygen should almost certainly be excluded from the non-ideality as in the MHR equations. I agree it was not done in the Isaacman-VanWertz and Aumont discussion of this approach, but I'm not sure that was actually correct (and, as I mentioned, the paper figures actually use the funcational group conversion approach) and should maybe be corrected in this present work. The author's response to my comment about re-working the equation does make sense though; in fact, it makes it easier to correct the issue with the non-ideality.**

Thank you.

**Line 459. Should be "either" instead of "neither"**

We revised the sentence to make it more clear, and thereby removing "neither" all in all

"Since none of the bN values were able to reproduce the data from different precursors for MHR, DMT, or Li, it is clear that simply adjusting the bN value would not resolve the discrepancy we observed between the measured and simulated VFR based on the chemical composition."

**Line 497. The authors state an intention to make the data public, so data access should be described in the Data Availability statement**

The data will be available through the Bolin Centre data base for everyone to download, we have updated the data availability description in the manuscript accordingly.

"The datasets will be published in the open access data base of the Bolin centre for climate research, available at https://bolin.su.se/data/."

**References**

Daumit, K. E., Kessler, S. H., and Kroll, J. H.: Average chemical properties and potential formation pathways of highly oxidized organic aerosol, Faraday Discussions, 165, 181-202, 10.1039/C3FD00045A, 2013.

Murray, K.K., Boyd, R.K, Eberlin, M.N., Langley, G.J., Li, L., and Naito, Y. Definitions of terms relating to mass spectrometry (IUPAC Recommendations 2013), Pure Appl. Chem., Vol 85, No. 7, pp. 1515-1609, 2013